# Log-Linear-Time Gaussian Processes
# Using Binary Tree Kernels

**Michael K. Cohen**
University of Oxford
`michael.cohen@eng.ox.ax.uk`

**Samuel Daulton**
University of Oxford, Meta
`sdaulton@meta.com`

**Michael A. Osborne**
University of Oxford
`mosb@robots.ox.ax.uk`

## Abstract

Gaussian processes (GPs) produce good probabilistic models of functions, but most GP kernels require $O((n + m)n^2)$ time, where $n$ is the number of data points and $m$ the number of predictive locations. We present a new kernel that allows for Gaussian process regression in $O((n + m) \log(n + m))$ time. Our "binary tree" kernel places all data points on the leaves of a binary tree, with the kernel depending only on the depth of the deepest common ancestor. We can store the resulting kernel matrix in $O(n)$ space in $O(n \log n)$ time, as a sum of sparse rank-one matrices, and approximately invert the kernel matrix in $O(n)$ time. Sparse GP methods also offer linear run time, but they predict less well than higher dimensional kernels. On a classic suite of regression tasks, we compare our kernel against Matérn, sparse, and sparse variational kernels. The binary tree GP assigns the highest likelihood to the test data on a plurality of datasets, usually achieves lower mean squared error than the sparse methods, and often ties or beats the Matérn GP. On large datasets, the binary tree GP is fastest, and much faster than a Matérn GP.

## 1 Introduction

Gaussian processes (GPs) can be used to perform regression with high-quality uncertainty estimates, but they are slow. Naïvely, GP regression requires $O(n^3 + n^2m)$ computation time and $O(n^2)$ computation space when predicting at $m$ locations given $n$ data points [28]. A kernel matrix of size $n \times n$ must be inverted (or Cholesky decomposed), and then $m$ matrix-vector multiplications must be done with that inverse matrix (or $m$ linear solves with the Cholesky factors). A few methods that we will discuss later achieve $O(n^2m)$ time complexity [25, 30].

With special kernels, GP regression can be faster and use less space. Inducing point methods, using $z$ inducing points, allow regression to be done in $O(z^2(n + m))$ time and in $O(z^2 + zn)$ space [21, 22, 23, 13]. We will discuss the details of these inducing point kernels later, but they are kernels in their own right, not just approximations to other kernels. Unfortunately, these kernels are low dimensional (having a $z$-dimensional Hilbert space), which limits the expressivity of the GP model.

We present a new kernel, the *binary tree kernel*, that also allows for GP regression in $O(n + m)$ space and $O((n + m) \log(n + m))$ time (both model fitting and prediction). The time and space complexity of our method is also linear in the depth of the binary tree, which is naïvely linear in the dimension of the data, although in practice we can increase the depth sublinearly. Training some kernel parameters takes time quadratic in the depth of the tree. The dimensionality of the binary tree kernel is exponential in the depth of the tree, making it much more expressive than an inducing points kernel. Whereas for an inducing points kernel, the runtime is quadratic in the dimension of the Hilbert space, for the binary tree kernel, it is only logarithmic—an exponential speedup.

A simple depiction of our kernel is shown in Figure 1, which we will define precisely in Section 3. First, we create a procedure for placing all data points on the leaves of a binary tree. Given the

binary tree, the kernel between two points depends only on the depth of the deepest common ancestor. Because very different tree structures are possible for the data, we can easily form an ensemble of diverse GP regression models. Figure 2 depicts a schematic sample from a binary tree kernel. Note how the posterior mean is piecewise flat, but the pieces can be small.

On a standard suite of benchmark regression tasks [25], we show that our kernel usually achieves better negative log likelihood (NLL) than state-of-the-art sparse methods and conjugate-gradient-based "exact" methods, at lower computational cost in the big-data regime.

There are not many limitations to using our kernel. The main limitation is that other kernels sometimes capture the relationships in the data better. We do not have a good procedure for understanding when data has more Matérn character or more binary tree character (except through running both and comparing training NLL). But

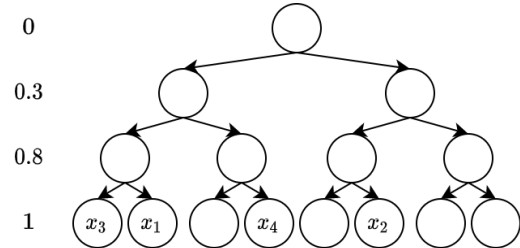

Figure 1: A binary tree kernel with four data points. In this example, $k(x_1, x_1) = 1$, $k(x_1, x_2) = 0$, $k(x_1, x_3) = 0.8$, and $k(x_1, x_4) = 0.3$.

given that the binary tree kernel usually outperforms the Matérn, we'll tentatively say the best first guess is that a new dataset has more binary tree character. One concrete limitation for some applications, like Bayesian Optimization, is that the posterior mean is piecewise-flat, so gradient-based heuristics for finding extrema would not work.

In contexts where a piecewise-flat posterior mean is suitable, we struggle to see when one would prefer a sparse or sparse variational GP to a binary tree kernel. The most thorough approach would be to run both and see which has a better training NLL, but if you had to pick one, the binary tree GP seems to be better performing and comparably fast. If minimizing mean-squared error is the objective, the Matérn kernel seems to do slightly better than the binary tree. If the dataset is small, and one needs a very fast prediction, a Matérn kernel may be the best option. But otherwise, if one cares about well-calibrated predictions, these initial results we present tentatively suggest using a binary tree kernel over the widely-used Matérn kernel.

The log-linear time and linear space complexity of the binary tree GP, with performance *exceeding* a "normal" kernel, could profoundly expand the viability of GP regression to larger datasets.

## 2 Preliminaries

Our problem setting is regression. Given a function $f : \mathcal{X} \to \mathbb{R}$, for some arbitrary set $\mathcal{X}$, we would like to predict $f(x)$ for various $x \in \mathcal{X}$. What we have are observations of $f(x)$ for various (other) $x \in \mathcal{X}$. Let $X \in \mathcal{X}^n$ be an $n$-tuple of elements of $\mathcal{X}$, and let $y \in \mathbb{R}^n$ be an $n$-tuple of real numbers, such that $y_i \sim f(X_i) + \mathcal{N}(0, \lambda)$, for $\lambda \in \mathbb{R}^{\geq 0}$. $X$ and $y$ comprise our training data.

With an $m$-tuple of test locations $X' \in \mathcal{X}^m$, let $y' \in \mathbb{R}^m$, with $y'_i = f(X'_i)$. $y'$ is the ground truth for the target locations. Given training data, we would like to produce a distribution over $\mathbb{R}$ for each target location $X'_i$, such that it assigns high marginal probability to the unknown $y'_i$. Alternatively, we sometimes would like to produce point estimates $\hat{y}'_i$ in order to minimize the squared error $(\hat{y}'_i - y'_i)^2$.

A GP prior over functions is defined by a mean function $m : \mathcal{X} \to \mathbb{R}$, and a kernel $k : \mathcal{X} \times \mathcal{X} \to \mathbb{R}$. The expected function value at a point $x$ is defined to be $m(x)$, and the covariance of the function values at two points $x_1$ and $x_2$ is defined to be $k(x_1, x_2)$. Let $K_{XX} \in \mathbb{R}^{n \times n}$ be the matrix of kernel

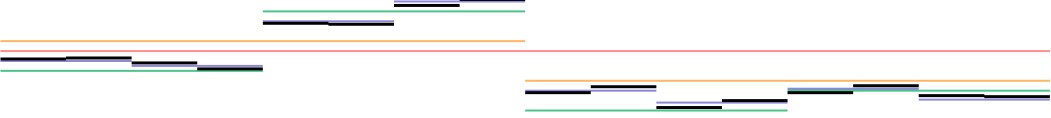

Figure 2: A schematic diagram of a function sampled from a binary tree kernel. The function is over the interval [0, 1], and points on the interval are placed onto the leaves of a depth-4 binary tree according to the first 4 bits of their binary expansion. The sampled function is in black. Purple represents the sample if the tree had depth 3, green depth 2, orange depth 1, and red depth 0.

values $(K_{XX})_{ij} = k(X_i, X_j)$, and let $m_X \in \mathbb{R}^n$ be the vector of mean values $(m_X)_i = m(X_i)$. For a GP to be well-defined, the kernel must be such that $K_{XX}$ is positive semidefinite for any $X \in \mathcal{X}^n$. For a point $x \in \mathcal{X}$, Let $K_{Xx} \in \mathbb{R}^n$ be the vector of kernel values: $(K_{Xx})_i = k(X_i, x)$, and let $K_{xX} = K_{Xx}^\top$. Let $\lambda \geq 0$ be the variance of observation noise. Let $\mu_x$ and $\sigma_x^2$ be the mean and variance of our posterior predictive distribution at $x$. Then, with $K_{XX}^{\lambda\text{inv}} = (K_{XX} + \lambda I)^{-1}$,

$$\mu_x := (y - m_X)^\top K_{XX}^{\lambda\text{inv}} K_{Xx} + m(x) \quad (1) \qquad \sigma_x^2 := k(x, x) - K_{xX} K_{XX}^{\lambda\text{inv}} K_{Xx} + \lambda. \quad (2)$$

See Williams and Rasmussen [28] for a derivation. We compute Equations 1 and 2 for all $x \in X'$.

# 3 Binary tree kernel

We now introduce the binary tree kernel. First, we encode our data points as binary strings. So we have $\mathcal{X} = \mathbb{B}^q$, where $\mathbb{B} = \{0, 1\}$, and $q \in \mathbb{N}$.

If $\mathcal{X} = \mathbb{R}^d$, we must map $\mathbb{R}^d \mapsto \mathbb{B}^q$. First, we rescale all points (training points and test points) to lie within the box $[0, 1]^d$. (If we have a stream of test points, and one lands outside of the box $[0, 1]^d$, we can either set $K_{xX}$ to $\mathbf{0}$ for that point or we rescale and retrain in $O(n \log n)$ time.) Then, for each $x \in [0, 1]^d$, for each dimension, we take the binary expansion up to some precision $p$, and for those $d \times p$ bits, we permute them using some fixed permutation. We call this permutation the bit order, and it is the same for all $x \in [0, 1]^d$. Note that now $q = dp$. See Figure 3 for an example. We optimize the bit order during training, and we can also form an ensemble of GPs using different bit orders.

For $x \in \mathbb{B}^q$, let $x^{\leq i}$ be the first $i$ bits of $x$. $[\![\text{expression}]\!]$ evaluates to 1, if expression is true, otherwise 0. We now define the kernel:

**Definition 1** (Binary Tree Kernel). *Given a weight vector $w \in \mathbb{R}^q$, with $w \succeq 0$ and $||w||_1 = 1$,*

$$k_w(x_1, x_2) = \sum_{i=1}^{q} w_i \left[\!\left[ x_1^{\leq i} = x_2^{\leq i} \right]\!\right]$$

So the more leading bits shared by $x_1$ and $x_2$, the larger the covariance between the function values. Consider, for example, points $x_1$ and $x_4$ from Figure 1, where $x_1$ is (left, left, right), and $x_4$ is (left, right, right); they share only the first leading "bit". We train the weight vector $w$ to maximize the likelihood of the training data.

Figure 3: Function from $[0, 1]^2 \to \mathbb{B}^8$.

**Proposition 1** (Positive Semidefiniteness). *For $X \in \mathcal{X}^n$, for $k = k_w$, $K_{XX} \succeq 0$.*

*Proof.* Let $s \in \bigcup_{i=1}^{q} \mathbb{B}^i$ be a binary string, and let $|s|$ be the length of $s$. Let $X_{[s]} \in \mathbb{R}^n$ with $(X_{[s]})_j = \left[\!\left[ X_j^{\leq |s|} = s \right]\!\right]$. $X_{[s]} X_{[s]}^\top$ is clearly positive semidefinite. Finally, $K_{XX} = \sum_{i=1}^{q} \sum_{s \in \mathbb{B}^i} w_i X_{[s]} X_{[s]}^\top$, and recall $w_i \geq 0$, so $K_{XX} \succeq 0$. $\qquad \square$

# 4 Sparse rank one sum representation

In order to do GP regression in $O(n)$ space and $O(n \log n)$ time, we develop a "Sparse Rank One Sum" representation of linear operators (SROS). This was developed separately from the very similar Hierarchical matrices [1], which we discuss below. In SROS form, linear transformation of a vector can be done in $O(n)$ time instead of $O(n^2)$. We will store our kernel matrix and inverse kernel matrix in SROS form. The proof of Proposition 1 exemplifies representing a matrix as the sum of sparse rank one matrices. Note that each $X_{[s]}$ is sparse—if $q$ is large, most $X_{[s]}$'s are the zero vector.

We now show how to interpret an SROS representation of an $n \times n$ matrix. Let $[n] = \{1, 2, ..., n\}$. For $r \in \mathbb{N}$, let $L : [r]^n \times [r]^n \times \mathbb{R}^n \times \mathbb{R}^n \to \mathbb{R}^{n \times n}$ construct a linear operator from four vectors.

**Definition 2** (Linear Operator from Simple SROS Representation). *Let $p, p' \in [r]^n$, and let $u, u' \in \mathbb{R}^n$. For $l \in [r]$, let $u^{p=l} \in \mathbb{R}^n$ be the vector where $u_j^{p=l} = u_j [\![ p_j = l ]\!]$, likewise for $u'$ and $p'$. Then:* $L(p, p', u, u') \mapsto \sum_{i=1}^{m} u^{p=i}(u')^{p'=i\top}$.

We depict Definition 2 in Figure 4. $p$ and $p'$ represent partitions over $n$ elements: all elements with the same integer value in the vector $p$ belong to the same partition. Note that $r$, the number of parts in the partition, need not exceed $n$, the number of elements being partitioned. If $p = p'$ (which is almost always the case for us) and the elements of $p$, $u$, and $u'$ were shuffled so that all elements in the same partition were next to each other, then $L(p, p', u, u')$ would be block diagonal. Note that $L(p, p', u, u')$ is not necessarily low rank. If $p$ is the finest possible partition, and $p = p'$, $L(p, p', u, u')$ is diagonal. SROS matrices can be thought of as a generalization of two types of matrix that are famously amenable to fast computation: rank one matrices (all points in the same partition) and diagonal matrices (each point in its own partition).

We now extend the definition of $L$ to allow for multiple $p$, $p'$, $u$, and $u'$ vectors.

**Definition 3** (Linear Operator from SROS Representation). *Let $L : [r]^{n \times q} \times [r]^{n \times q} \times \mathbb{R}^{n \times q} \times \mathbb{R}^{n \times q} \to \mathbb{R}^{n \times n}$. Let $P, P' \in [r]^{n \times q}$, and let $U, U' \in \mathbb{R}^{n \times q}$. Let $P_{:,i}$, $U_{:,i}$, etc. be the $i^{th}$ columns of the respective arrays. Then: $L(P, P', U, U') \mapsto \sum_{i=1}^{q} L(P_{:,i}, P'_{:,i}, U_{:,i}, U'_{:,i})$.*

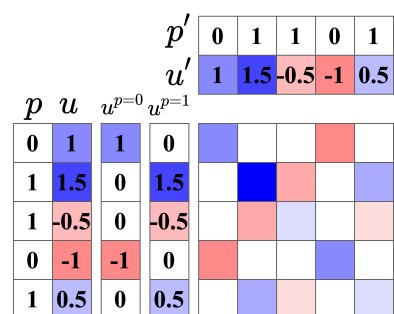

Figure 4: A matrix in standard form constructed from a matrix in SROS form. The large square depicts the matrix $L(p, p', u, u') \in \mathbb{R}^{5 \times 5}$ with elements colored by value. See $u^{p=0}$ for a color legend.

Algorithm 1 performs linear transformation of a vector using SROS representation in $O(nq)$ time.

---

**Algorithm 1** Linear Transformation with SROS Linear Operator. This can be vectorized on a Graphical Processing Unit (GPU), using e.g. `torch.Tensor.index_add_` for Line 5 and non-slice indexing for Line 6 [19]. Slight restructuring allows vectorization over $[q]$ as well.

---

**Require:** $P, P' \in [r]^{n \times q}, U, U' \in \mathbb{R}^{n \times q}, x \in \mathbb{R}^n$
**Ensure:** $y = L(P, P', U, U')x$
1: $y \leftarrow \mathbf{0} \in \mathbb{R}^n$
2: **for** $i \in [q]$ **do** $\qquad\qquad\qquad\qquad\qquad\qquad\qquad\qquad\qquad\qquad$ ▷ $O(nq)$ time
3: $\quad p, p', u, u' \leftarrow P_{:,i}, P'_{:,i}, U_{:,i}, U'_{:,i}$
4: $\quad z \leftarrow \mathbf{0} \in \mathbb{R}^r \qquad\qquad\qquad\qquad$ ▷ $z_l$ will store the dot product $((u')^{p'=l})^\top x^{p'=l}$
5: $\quad$ **for** $j \in [n]$ **do** $z_{p'_j} \leftarrow z_{p'_j} + u'_j x_j \qquad\qquad\qquad\qquad\qquad\qquad$ ▷ $O(n)$ time
6: $\quad$ **for** $j \in [n]$ **do** $y_j \leftarrow y_j + z_{p_j} u_j \qquad\qquad\qquad\qquad\qquad\qquad$ ▷ $O(n)$ time
$\quad$ **return** $y$

---

We now discuss how to approximately invert a certain kind of symmetric SROS matrix, but our methods could be extended to asymmetric matrices. First, we define a partial ordering over partitions. For two partitions $p, p'$, we say $p' \leq p$ if $p'$ is finer than or equal to $p$; that is, $p'_j = p'_{j'} \implies p_j = p_{j'}$. Using that partial ordering, a symmetric SROS matrix can be approximately inverted efficiently if for all $1 \leq i, i' \leq q$, $P_{:,i} \leq P_{:,i'}$ or $P_{:,i'} \leq P_{:,i}$. As the reader may have recognized, our kernel matrix $K_{XX}$ can be written as an SROS matrix with this property.

We will write symmetric SROS matrices in a slightly more convenient form. All $(u')^{p=l}$ must be a constant times $u^{p=l}$. We will store these constants in an array $C$. Let $L(P, C, U)$ be shorthand for $L(P, P, U, C \odot U)$, where $\odot$ denotes element-wise multiplication. For $L(P, C, U)$ to be symmetric, it must be the case that $P_{ji} = P_{j'i} \implies C_{ji} = C_{j'i}$. Then, all elements of $U$ corresponding to a given $u^{p=l}$ are multiplied by the same constant. We now present an algorithm for calculating $(L(P, C, U) + \lambda I)^{-1}$, for $\lambda \neq 0$, which is an approximate inversion of $L(P, C, U)$. We have not yet analyzed numerical sensitivity for $\lambda \to 0$, but we conjecture that all floating point numbers involved need to be stored to at least $\log_2(1/\lambda)$ bits. Without loss of generality, let $\lambda = 1$, and note $(L(P, C, U) + \lambda I)^{-1} = \lambda^{-1}(L(P, \lambda^{-1}C, U) + I)^{-1}$.

By assumption, all columns of $P$ are comparable with respect to the partial ordering above, so we can reorder the columns of $P$ such that $P_{:,i} \geq P_{:,j}$ for $i < j$. The key identity that we use to develop our fast inversion algorithm is the Sherman–Morrison Formula:

$$(A + cuu^\top)^{-1} = A^{-1} - \frac{A^{-1}uu^\top A^{-1}}{c^{-1} + u^\top A^{-1}u} \tag{3}$$

Starting with $A = I$, we add the sparse rank one matrices iteratively, from the finest partition to the coarsest one, updating $A^{-1}$ as we go. We represent $(L(P, C, U) + I)^{-1}$ in the form $I + L(P, C', U')$, so we write an algorithm that returns $C'$ and $U'$. We can also quickly calculate $\log |L(P, C, U) + I|$ at the same time, using the matrix determinant lemma: $|A + cuu^\top| = (1 + cu^\top A^{-1}u)|A|$.

**Theorem 1** (Fast Inversion). *For $P \in [r]^{n \times q}$ and $C, U \in \mathbb{R}^{n \times q}$, if $P_{:,i} \overset{\text{(is coarser than)}}{\geq} P_{:,j}$ for $i < j$, then there exists $C', U' \in \mathbb{R}^{n \times q}$, such that $(L(P, C, U) + I)^{-1} = I + L(P, C', U')$. There exists an algorithm for computing $C'$ and $U'$ that takes $O(nq^2)$ time.*

*Proof.* For $X \in \mathbb{R}^{n \times q}$, let $X_{:,i+1:q} \in \mathbb{R}^{n \times (q-i)}$ be columns $i + 1$ through $q$ of matrix $X$ (inclusive). Let $A_i = I + L(P_{:,i+1:q}, C_{:,i+1:q}, U_{:,i+1:q})$, and $A_q = I$. Now suppose $A_i^{-1}$ can be written as $I + L(P_{:,i+1:q}, C'_{:,i+1:q}, U'_{:,i+1:q})$ for some $C'$ and $U'$. For the base case of $i = q$, this holds trivially. We show it also holds for $i - 1$, and we can compute $C'_{:,i:q}, U'_{:,i:q}$ in $O(n(q - i))$ time. Let $p = P_{:,i}$, $u = U_{:,i}$, and $c = C_{:,i}$. Consider $u^{p=l}$, where each element is zero unless the corresponding element of $p$ equals $l$. What do we know about the product $A_i^{-1}u^{p=l}$ (as seen in Equation 3)?

Because the columns of $P$ go from coarser partitions to finer ones, all of the vectors generating the sparse rank one components of $L(P_{:,i+1:q}, C'_{:,i+1:q}, U'_{:,i+1:q})$ are from partitions that are equal to or finer than $p$. Thus, they are either zero everywhere $u^{p=l}$ is zero, or zero everywhere $u^{p=l}$ is nonzero. Vectors $v$ of the second kind can be ignored, as $cvv^\top u^{p=l} = 0$. Thus, when multiplying $L(P_{:,i+1:q}, C'_{:,i+1:q}, U'_{:,i+1:q})$ by $u^{p=l}$, the only relevant vectors are filled with zeros except where the corresponding element of $p$ equals $l$. So we can get rid of those rows of $P_{:,i+1:q}$, $C'_{:,i+1:q}$, and $U'_{:,i+1:q}$. Suppose there are $n_l$ elements of $p$ that equal $l$. Then $L(P_{:,i+1:q}, C'_{:,i+1:q}, U'_{:,i+1:q})u^{p=l}$ involves $n_l$ rows, and can be computed in $O(n_l(q - i))$ time. Moreover, this product, which we'll call $(u')^{p=l}$, is only nonzero when the corresponding element of $p$ equals $l$, so it has the same sparsity pattern as $u^{p=l}$. The other component of $A_i^{-1}$ is the identity matrix, and $Iu^{p=l}$ clearly has the same sparsity as $u^{p=l}$. Thus, returning to Equation 3, when we add $u^{p=l}(u^{p=l})^\top$ to $A_i$, we update $A_i^{-1}$ with an outer product of vectors whose sparsity pattern is the same as that of $u^{p=l}$.

For each $l$, $A_i^{-1}$ need not be updated with each $u^{p=l}$ one at a time. For $l \neq l'$, $u^{p=l}$ and $u^{p=l'}$ are nonzero at separate indices, so $u^{p=l}$ and $(u')^{p=l'}$ are nonzero at separate indices, so the extra component of $A_i^{-1}$ that appears after the $u^{p=l'}$ update is irrelevant to the $u^{p=l}$ update, because $(u^{p=l})^\top (u')^{p=l'} = 0$. Since the $u^{p=l}$ update takes $O(n_l(q - i))$ time, all of them together take $O(\sum_l n_l(q - i))$ time, which equals $O(n(q - i))$ time. Calculating an element of $c'$ only involves computing the denominator in Equation 3, using a matrix-vector product already computed. So we can write $C'_{:,i:q}$ and $U'_{:,i:q}$ by adding a preceding column to $C'_{:,i+1:q}$ and $U'_{:,i+1:q}$, using the same partition $p$, and it takes $O(n(q - i))$ time.

Following the induction down to $i = 0$, we have $(L(P, C, U) + I)^{-1} = I + L(P, C', U')$, and a total time of $O(nq^2)$. $\qquad\square$

Algorithm 2 also performs approximate inversion, which we prove in Appendix A. It differs slightly from the algorithm in the proof, but can take full advantage of a GPU speedup. In the setting where all columns of $U$ are identical, observe that in Lines 10 and 11, the same computation is repeated for all $k \in [i]$. Indeed, in this setting, this block of code can be modified to run in $O(n)$ time rather than $O(ni)$, making the whole algorithm run in $O(nq)$ time, as shown in Proposition 4 in Appendix A.

A Hierarchical matrix is a matrix which is either represented as a low-rank matrix or as a $2 \times 2$ block matrix of Hierarchical matrices [1]. In our SROS format, many of the sparse rank one matrices overlap, whereas in a Hierarchical matrix, the low-rank matrices do not overlap, and converting an SROS matrix into a Hierarchical matrix would typically be inefficient. Hierarchical matrices admit approximate inversion in $O(na^2 \log^2 n)$ time, where $a$ is the maximum rank of the component submatrices [11]. However, this is not an approximation in a technical sense, as there is no error bound. At many successive steps in the algorithm, a rank $2a$ matrix is approximated by a rank $a$

**Algorithm 2** Inverse and determinant of $I+$ SROS Linear Operator. Lines 5 through 11 can all be easily vectorized on a GPU. Lines 5 and 10 require `torch.Tensor.index_add_` or equivalent, and lines 6 and 11 require non-slice indexing, which are not quite as fast as some GPU operations.

---

**Require:** $P \in [r]^{n \times q}, C, U \in \mathbb{R}^{n \times q}$
**Ensure:** $I + L(P, C', U') = (I + L(P, C, U))^{-1}$; $x = \log|I + L(P, C, U)|$
1:   $x, C', U' \leftarrow 0, \mathbf{0} \in \mathbb{R}^{n \times q}, U$
2:   **for** $i \in (q, q-1, ..., 1)$ **do**             $\triangleright O(nq^2)$ time
3:    $p, c, u, u' \leftarrow P_{:,i}, C_{:,i}, U_{:,i}, U'_{:,i}$
4:    $z \leftarrow \mathbf{0} \in \mathbb{R}^r$     $\triangleright z_l$ will store $c^{(l)}((u')^{p=l})^\top u^{p=l}$, where $c^{(l)} = c_k$ if $p_k = l$
5:    **for** $j \in [n]$ **do** $z_{p_j} \leftarrow z_{p_j} + c_j u'_j u_j$         $\triangleright O(n)$ time
6:    **for** $j \in [n]$ **do** $C'_{ji} \leftarrow -c_j/(1 + z_{p_j})$       $\triangleright O(n)$ time
7:    **for** $l \in [r]$ **do** $x \leftarrow x + \log(1 + z_l)$     $\triangleright O(n)$ time because $r \leq n$
8:    **if** $i > 0$ **then**
9:     $y \leftarrow \mathbf{0} \in \mathbb{R}^{n \times i}$             $\triangleright O(ni)$ time
10:     **for** $j, k \in [n] \times [i-1]$ **do** $y_{p_j k} \leftarrow y_{p_j k} + u'_j U_{jk}$    $\triangleright O(ni)$ time
11:     **for** $j, k \in [n] \times [i-1]$ **do** $U'_{jk} \leftarrow U'_{jk} + C'_{ji} u'_j y_{p_j k}$   $\triangleright O(ni)$ time
  **return** $C', U', x$

---

matrix [10]; to our knowledge there is no analysis of how resulting errors might cascade. After converting an SROS matrix to hierarchical form, this rough inversion would take $O(nq^2 \log^2 n)$ time.

## 5   Binary tree Gaussian process

We now show that our kernel matrix $K_{XX}$ can be written in SROS form, with $P$ containing successively finer partitions. Thus, $K_{XX}$ can be approximately inverted quickly, for use in Equations 1 and 2. Next, we'll show that we can efficiently optimize the log likelihood of the training data by tuning the weight vector $w$ along with the bit order. The log likelihood can be calculated in $O(nq \log n)$ time and then the gradient w.r.t. $w$ in $O(nq^2)$ time.

Recall from the proof of Proposition 1: $K_{XX} = \sum_{i=1}^{q} \sum_{s \in \mathbb{B}^i} w_i X_{[s]} X_{[s]}^\top$, where $X_{[s]} \in \mathbb{R}^n$ with $(X_{[s]})_j = \left[\!\!\left[ X_j^{\leq |s|} = s \right]\!\!\right]$. So we will set $P_{:,i}$, $C_{:,i}$, and $U_{:,i}$, so that $L(P_{:,i}, C_{:,i}, U_{:,i}) = \sum_{s \in \mathbb{B}^i} w_i X_{[s]} X_{[s]}^\top$. Let $P_{:,i}$ partition the set of points $X$ so that points are in the same partition if the first $i$ bits match. Now, requiring the first $i+1$ bits to match is a stricter criterion than requiring the first $i$ bits to match, so the $P_{:,i}$ grow successively finer. For any piece of the partition where the first $i$ bits of the constituent points equals the bitstring $s$, the corresponding sparse rank one component of $K_{XX}$ is $w_i X_{[s]} X_{[s]}^\top$. So let $U_{:,i} = \mathbf{1}^n$, and let $C_{:,i} = w_i \mathbf{1}^n$.

**Proposition 2** (SROS Form Kernel). $K_{XX} = L(P, C, U)$, *as defined above.*

This follows immediately from the definitions. To compute these partitions $P_{:,i}$, we sort $X$, which is a set of bit strings. And then we can easily compute which points have the same first $i$ bits. This all takes $O(nq \log n)$ time. Now note that $U_{:,i} = U_{:,i'}$ for all $i, i'$, so $(K_{XX} + \lambda I)^{-1}$ and $|K_{XX} + \lambda I|$ can be computed in $O(nq)$ time, rather than $O(nq^2)$.

The training negative log likelihood of a GP is that of the corresponding multivariate Gaussian on the training data. So: $\text{NLL}(w) = \frac{1}{2}\left(y^\top (K_{XX}(w) + \lambda I)^{-1} y + \log|K_{XX}(w) + \lambda I| + n \log(2\pi)\right)$. This can be computed in $O(nq)$ time, since matrix-vector multiplication takes $O(nq)$ time for a matrix in SROS form. So if the bit order is unchanged, an optimization step can be done in $O(nq)$ time, and if the data needs to be resorted, then in $O(nq \log n)$ time. On the largest dataset we tested (House Electric), with $n \approx 1.3$ million and $q = 88$, sorting the data and computing $P$ takes about 0.96 seconds on a GPU, and then calculating the negative log likelihood takes about another 1.08 seconds. We show in Appendix B how to compute $\nabla_w \text{NLL}$ in $O(nq^2)$ time.

To optimize the bit order and weight vector at the same time, we represent both with a single parameter vector $\theta \in \mathbb{R}_+^q$, with $||\theta||_\infty = 1$. To get the bit order from $\theta$, we start with a default bit order and permute the bit order according to a permutation that would sort $\theta$ in descending order. To get the

**Algorithm 3** GP Regression with a binary tree kernel.

---

**Require:** $X \in \mathbb{B}^{n \times q}$, $y \in \mathbb{R}^n$, $X' \in \mathbb{B}^{m \times q}$, $w \in \mathbb{R}^q$, $\lambda \in \mathbb{R}^+$
**Ensure:** $\mu_{X'}$ and $\sigma^2_{X'}$ are the predictive means and variances at $X'$, and nll the training negative log likelihood.

1: $\tilde{X} \leftarrow X \circ X'$
2: $\tilde{X}^\uparrow$, perm $\leftarrow$ Sort$(\tilde{X})$          ▷ The rows of $X$ are sorted lexically from leading bit to trailing bit. $O((n+m)q\log(n+m))$ time.
3: **for** $j, i \in [n+m] \times [q]$ **do** $\tilde{P}^\uparrow_{ji} \leftarrow$ #of unique rows in $X^\uparrow_{1:j,1:i}$
4:          ▷ $M_{1:j,1:i}$ is the first $j$ rows and $i$ columns of $M$. $O((n+m)q)$ time.
5: $\tilde{P} \leftarrow$ perm$^{-1}(P^\uparrow)$          ▷ This "unsorts" the input. $O((n+m)q)$ time.
6: $P, P' \leftarrow \tilde{P}_{1:n}, \tilde{P}_{n+1:n+m}$
7: $U, U', \tilde{U} \leftarrow \mathbf{1}^{n \times q}, \mathbf{1}^{m \times q}, \mathbf{1}^{(n+m) \times q}$
8: $C, \tilde{C} \leftarrow \mathbf{1}^n w^T, \mathbf{1}^{n+m} w^T$
9: $C^{-1}_\lambda, U^{-1}, \text{logdet}_\lambda \leftarrow$ Invert$(P, \lambda^{-1}C, U)$    ▷ Uses Algorithm 2. Speedup to $O(nq)$ time because columns of $U$ are identical.
10: $C^{-1}, \text{logdet} \leftarrow \lambda^{-1}C^{-1}_\lambda, \text{logdet}_\lambda + n\log(\lambda)$
11: $z \leftarrow$ LinTransform$(P, P, U^{-1}, C^{-1} \odot U^{-1}, y) + \lambda^{-1}y$    ▷ Uses Algorithm 1 to compute the Woodbury vector. $O(nq)$ time.
12: $\mu_{X'} \leftarrow$ LinTransform$(P', P, U', C \odot U, z)$          ▷ $O((n+m)q)$ time.
13: nll $\leftarrow (y^\top z + \text{logdet} + n\log(2\pi))/2$
14: $\tilde{C}^{\text{prec}}, \tilde{U}^{\text{prec}} \leftarrow$ Invert$(\tilde{P}, \lambda^{-1}\tilde{C}, \lambda^{-1}\tilde{U})$          ▷ $O((n+m)q)$ time.
15: $C^{\text{prec}}, U^{\text{prec}} \leftarrow \tilde{C}^{\text{prec}}_{n+1:n+m}, \tilde{U}^{\text{prec}}_{n+1:n+m}$
16: $C^{\text{cov}}, U^{\text{cov}} \leftarrow$ Invert$(P', C^{\text{prec}}, U^{\text{prec}})$    ▷ $O(mq^2)$ time; extra factor of $q$ because columns of $U^{\text{prec}}$ are not identical.
17: $\sigma^2_{X'} \leftarrow \lambda(\mathbf{1}^m + \text{SumEachRow}(C^{\text{cov}} \odot U^{\text{cov}} \odot U^{\text{cov}}))$    ▷ $O(mq)$ time.
18: **return** $\mu_{X'}, \sigma^2_{X'}$, nll

---

weight vector, we sort $\theta$ in descending order, add a $0$ at the end, and compute the differences between adjacent elements. When there are ties in the elements of $\theta$, the choice of bit order does not affect the negative log likelihood (or the kernel at all) because the relevant associated weight is $0$. The negative log likelihood is continuous with respect to $\theta$, and when all values of $\theta$ are unique, it is differentiable with respect to $\theta$. Letting $\theta = e^\phi / ||e^\phi||_\infty$, we minimize loss w.r.t. $\phi$ using BFGS [8].

To calculate the predictive mean at a list of predictive locations $X'$, we first multiply $y$ by $(K_{XX} + \lambda I)^{-1}$, and then we multiply that vector by $K_{XX'}$. We obtain both $K_{XX}$ and $K_{XX'}$ in SROS form as follows. Let $\tilde{X} = X \circ X'$ be the concatenation of the two tuples, now an $(n+m)$-tuple. Writing $K_{\tilde{X}\tilde{X}} = L(\tilde{P}, \tilde{C}, \tilde{U})$, the arrays on the r.h.s. can be computed in $O((n+m)q\log(n+m))$ time. Then, with $P$, $C$, and $U$ being the first $n$ rows of $\tilde{P}, \tilde{C}, \tilde{U}$, $K_{XX} = L(P, C, U)$. And letting $P''$ and $U''$ be the last $m$ rows, $K_{XX'} = L(P, P'', C \odot U, U'')$. Thus, the predictive mean $\mu_x$ from Equation 1 can be computed at $m$ locations in $O((n+m)q\log(n+m))$ time.

The predictive covariance matrix, which extends the predictive variance from Equation 2, is calculated $\Sigma_{X'} = K_{X'X'} + \lambda I_m - K_{X'X}(K_{XX} + \lambda I_n)^{-1}K_{XX'} = (K_{\tilde{X}\tilde{X}} + \lambda I_{m+n})/K_{XX}$, where $/$ denotes the Schur complement. From a property of block matrix inversion, the last $m$ columns of the last $m$ rows of $(K_{\tilde{X}\tilde{X}} + \lambda I)^{-1}$ equals $((K_{\tilde{X}\tilde{X}} + \lambda I_{m+n})/K_{XX})^{-1}$. So we get the predictive precision matrix in $O((n+m)q\log(n+m))$ time by inverting $K_{\tilde{X}\tilde{X}} + \lambda I$ and taking the bottom right $m \times m$ block. Then, we get the predictive covariance matrix by inverting that. This takes $O(mq^2)$ time, since it does not have the property of all the columns of $U$ being equal. If we only want the diagonal elements of an SROS matrix (the independent predictive variances in this case), we can simply sum the rows of $C \odot U \odot U$ in $O(mq)$ time. Thus, in total, computing the independent predictive variances requires $O((n+m)q\log(n+m) + mq^2)$ time. See Algorithm 3.

## 6 Related Work

All existing kernels of which we are aware for linear time GP regression on unstructured data involve

inducing points (related to the Nyström approximation [27]) or inducing frequencies. For a given set of inducing points $Z$, for some base kernel $k$, the inducing point kernel (in its most basic form) is the following, although subtle variants exist: $k^Z(x, x') = K_{xZ}K_{ZZ}^{-1}K_{Zx'}$ [21].

Sparse Gaussian Process Regression (SGPR) involves selecting $Z$, and then using $k^Z$ (or a variant). Notably, $K_{XX}^Z = K_{XZ}K_{ZZ}^{-1}K_{ZX}$ is low rank, providing computational efficiency. The predictive mean and covariance have compact form, with observational noise $\lambda$: $\mu_x(Z) = K_{xZ}(\lambda K_{ZZ} + k_{ZX}k_{XZ})^{-1}K_{ZX}y$ and $\sigma_{xx'}^2(Z) = K_{xZ}(K_{ZZ} + \lambda^{-1}k_{ZX}k_{XZ})^{-1}K_{Zx'}$.

Titsias's [24] sparse variational kernel is also low rank and uses inducing points. The sparse variational GP (SVGP) is constructed as the solution to a variational inference problem. It depends on inducing points $Z$, data points $X$, and observed function values $y$. We have focused on Gaussian processes with 0 mean, but the SVGP method uses a nonzero prior mean along with a kernel: $m^{\mathrm{SVGP}}(x) = \mu_x(Z)$ and $k^{\mathrm{SVGP}}(x, x') = k(x, x') - K_{xZ}K_{ZZ}^{-1}K_{Zx'} + \sigma_{xx'}^2(Z)$. Given the dependence on $X$ and $y$, this is not a true probability distribution over function space. The variational problem underlying this kernel also provides guidance in how to select the inducing points $Z$. For further discussion of the kernel underlying the SVGP method, see Wild et al. [26].

An inducing point kernel with $z$ inducing points produces a $z$-dimensional reproducing kernel Hilbert space (RKHS). The dimensionality of the RKHS relates to the expressivity of the kernel. Whereas an inducing point method buys a $z$-dimensional RKHS for the price of $O(z^2 n)$ time and $O(zn)$ space, the binary tree kernel produces a $2^q$-dimensional RKHS in $O(qn)$ time and space—an exponential improvement. (Observe that we can find $2^q$ linearly independent functions of the form $k(\cdot, x)$—one for each of the $2^q$ leaves $x$ might belong to.) Wilson and Nickisch [29] develop a method for speeding up inducing point methods significantly, especially in low-dimensional settings.

Lázaro-Gredilla et al. [16] propose an inducing frequencies kernel: given a set of $m$ inducing vectors $s_i$, $k(x, x') = 1/m \sum_{i=1}^{m} \cos(2\pi s_i^\top (x - x'))$. Dutordoir et al. [6] propose an inducing frequencies kernel for low dimensional data, in which $k(x, x')$ is a special function of $x^\top x'$.

On one-dimensional data, filtering/smoothing methods perform Bayesian inference over functions in $O(n)$ time [14, 12]. A few non-$O(n)$ methods bear mentioning. We are not the first to consider a kernel over points on the leaves of a tree [18, 17] or on the leaves of multiple trees [7], but their methods take $O(n^3)$ time. On certain kinds of structured data, Toeplitz solvers achieve $O(n^2)$ time complexity [30]. Cutajar et al.'s [3], Wang et al.'s [25], and others' use of a conjugate gradients solver to replace inversion/factorization has unclear time complexity between $O(n^2)$ and $O(n^3)$, depending on the kernel matrix spectrum. [1]

## 7  Experiments

In Table 1, we compare our binary tree kernel and a binary tree ensemble (see Appendix C for details on the ensemble) against three baseline methods: exact GP regression using a Matérn kernel, sparse Gaussian process regression (SGPR) [23], and a stochastic variational Gaussian process (SVGP) [13]. We evaluate our method on the same open-access UCI datasets [4] as Wang et al. [25], using their same training, validation, and test partitions, and we compare against the baseline results they report. For the binary tree (BT) kernels, we use $p = \min(8, \lfloor 150/d \rfloor + 1)$, and recall $q = pd$. We set $\lambda = 1/n$. We train the bit order and weights to minimize training NLL. For the binary tree ensemble (BTE), we use 20 kernels. For the Matérn kernel, we use Blackbox Matrix-Matrix multiplication (BBMM) [9], which uses the conjugate gradients method to calculate matrix-vector products with $(K_{XX} + \lambda I)^{-1}$. SGPR uses 512 data points and SVGP uses 1,024 inducing points. We report the mean and two standard errors across 3 replications with different dataset splits. For further experimental details, see Appendix C. BTE achieves the best NLL on 6/12 datasets, and best RMSE on 5/12 datasets (including some ties). Out of the 4 largest datasets, BT/BTE is fastest on 3. The run

---

[1]The conjugate gradients method takes $O(n^2 \sqrt{\kappa})$ time, where $\kappa$ is the condition number of the kernel matrix. Poggio et al. [20] say "claims about the condition number of a random matrix A should also apply to kernel matrices with random data." If they mean a Wishart random matrix (which it should be if, e.g., $k(x, x') = x^\top x'$), that would be the square of the condition number of the corresponding Gaussian random matrix, which grows as $O(n)$ [2]. Putting it all together, we get $O(n^3)$ for conjugate gradients. We don't know how quickly preconditioning can reduce the condition number.

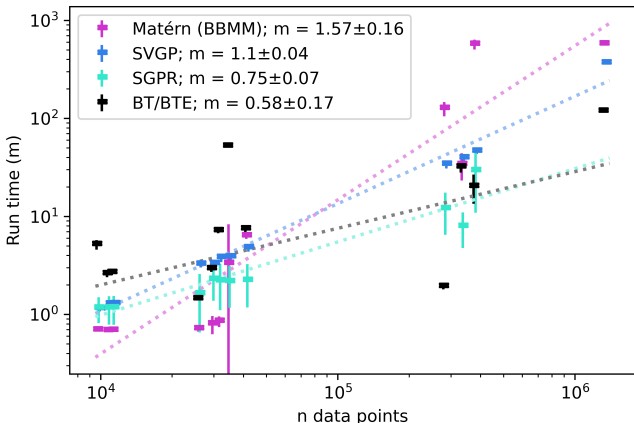

Figure 5: Run times given dataset size. For BT, the trendline is calculated controlling for $\log(q)$ with affine regression, and then setting $q = 150$. The slope w.r.t. $\log(q)$ is $2.82 \pm 1.06$. Theoretically, all slopes are too low except for that of SVGP, presumably because of overhead in the small-data regime.

times are plotted in Figure 5. The code is available at `https://github.com/mkc1000/btgp` and `https://tinyurl.com/btgp-colab`.

BT performs noticeably worse than BTE for test NLL on CTSlice due to over-fitting. There are enough degrees of freedom when optimizing the bit order ($d = 385$) that BT kernel can over-fit to the training data. The ensemble over multiple bit orders is much more robust.

| Dataset | $n$ | $d$ | BTE | BT | Matérn (BBMM) | SGPR | SVGP |
|---|---|---|---|---|---|---|---|
| PoleTele | 9,600 | 26 | $\mathbf{-0.625} \pm 0.035$ | $-0.490 \pm 0.040$ | $-0.180 \pm 0.036$ | $-0.094 \pm 0.008$ | $-0.001 \pm 0.008$ |
| Elevators | 10,623 | 18 | $0.649 \pm 0.032$ | $0.646 \pm 0.023$ | $0.619 \pm 0.054$ | $0.580 \pm 0.060$ | $\mathbf{0.519} \pm 0.022$ |
| Bike | 11,122 | 17 | $\mathbf{-0.708} \pm 0.433$ | $\mathbf{-0.806} \pm 0.273$ | $0.119 \pm 0.044$ | $0.291 \pm 0.032$ | $0.272 \pm 0.018$ |
| Kin40k | 25,600 | 8 | $0.869 \pm 0.004$ | $0.881 \pm 0.008$ | $\mathbf{-0.258} \pm 0.084$ | $0.087 \pm 0.067$ | $0.236 \pm 0.077$ |
| Protein | 29,267 | 9 | $\mathbf{0.781} \pm 0.023$ | $0.845 \pm 0.026$ | $1.018 \pm 0.056$ | $0.970 \pm 0.010$ | $1.035 \pm 0.006$ |
| KeggDir | 31,248 | 20 | $-1.031 \pm 0.020$ | $-1.029 \pm 0.021$ | $-0.199 \pm 0.381$ | $\mathbf{-1.123} \pm 0.016$ | $-0.940 \pm 0.020$ |
| CTslice | 34,240 | 385 | $\mathbf{-2.527} \pm 0.147$ | $-1.092 \pm 0.147$ | $-0.894 \pm 0.188$ | $-0.073 \pm 0.097$ | $1.422 \pm 0.005$ |
| KEGGU | 40,708 | 27 | $-0.667 \pm 0.007$ | $-0.667 \pm 0.007$ | $-0.419 \pm 0.027$ | $\mathbf{-0.984} \pm 0.012$ | $-0.666 \pm 0.007$ |
| 3DRoad | 278,319 | 3 | $\mathbf{-0.251} \pm 0.009$ | $\mathbf{-0.252} \pm 0.006$ | $0.909 \pm 0.001$ | $0.943 \pm 0.002$ | $0.697 \pm 0.002$ |
| Song | 329,820 | 90 | $1.330 \pm 0.003$ | $1.331 \pm 0.003$ | $\mathbf{1.206} \pm 0.024$ | $1.213 \pm 0.003$ | $1.417 \pm 0.000$ |
| Buzz | 373,280 | 77 | $1.198 \pm 0.003$ | $1.198 \pm 0.003$ | $0.267 \pm 0.028$ | $\mathbf{0.106} \pm 0.008$ | $0.224 \pm 0.050$ |
| HouseElec | 1,311,539 | 11 | $\mathbf{-2.569} \pm 0.006$ | $-2.492 \pm 0.012$ | $-0.152 \pm 0.001$ | — | $-1.010 \pm 0.039$ |
| PoleTele | 9,600 | 26 | $\mathbf{0.154} \pm 0.006$ | $0.161 \pm 0.004$ | $\mathbf{0.151} \pm 0.012$ | $0.217 \pm 0.002$ | $0.215 \pm 0.002$ |
| Elevators | 10,623 | 18 | $0.478 \pm 0.021$ | $0.476 \pm 0.018$ | $\mathbf{0.394} \pm 0.006$ | $0.437 \pm 0.018$ | $0.399 \pm 0.009$ |
| Bike | 11,122 | 17 | $0.118 \pm 0.057$ | $\mathbf{0.103} \pm 0.029$ | $0.220 \pm 0.002$ | $0.362 \pm 0.004$ | $0.303 \pm 0.004$ |
| Kin40k | 25,600 | 8 | $0.580 \pm 0.003$ | $0.587 \pm 0.006$ | $\mathbf{0.099} \pm 0.001$ | $0.273 \pm 0.025$ | $0.268 \pm 0.022$ |
| Protein | 29,267 | 9 | $0.608 \pm 0.008$ | $0.623 \pm 0.011$ | $\mathbf{0.536} \pm 0.012$ | $0.656 \pm 0.010$ | $0.668 \pm 0.005$ |
| KeggDir | 31,248 | 20 | $\mathbf{0.086} \pm 0.003$ | $\mathbf{0.086} \pm 0.003$ | $\mathbf{0.086} \pm 0.005$ | $0.104 \pm 0.003$ | $0.096 \pm 0.001$ |
| CTslice | 34,240 | 385 | $\mathbf{0.116} \pm 0.009$ | $0.132 \pm 0.009$ | $0.262 \pm 0.448$ | $0.218 \pm 0.011$ | $1.003 \pm 0.005$ |
| KEGGU | 40,708 | 27 | $0.120 \pm 0.001$ | $0.121 \pm 0.001$ | $\mathbf{0.118} \pm 0.000$ | $0.130 \pm 0.001$ | $0.124 \pm 0.002$ |
| 3DRoad | 278,319 | 3 | $0.187 \pm 0.002$ | $0.186 \pm 0.001$ | $\mathbf{0.101} \pm 0.007$ | $0.661 \pm 0.010$ | $0.481 \pm 0.002$ |
| Song | 329,820 | 90 | $0.914 \pm 0.003$ | $0.916 \pm 0.003$ | $\mathbf{0.807} \pm 0.024$ | $0.803 \pm 0.002$ | $0.998 \pm 0.000$ |
| Buzz | 373,280 | 77 | $0.801 \pm 0.002$ | $0.801 \pm 0.002$ | $\mathbf{0.288} \pm 0.018$ | $0.300 \pm 0.004$ | $\mathbf{0.304} \pm 0.012$ |
| HouseElec | 1,311,539 | 11 | $\mathbf{0.029} \pm 0.001$ | $\mathbf{0.029} \pm 0.001$ | $0.055 \pm 0.000$ | — | $0.084 \pm 0.005$ |
| PoleTele | 9,600 | 26 | $5.16 \pm 0.58$ | | $\mathbf{0.69} \pm 0.018$ | $1.16 \pm 0.34$ | $1.15 \pm 0.068$ |
| Elevators | 10,623 | 18 | $2.6 \pm 0.19$ | | $\mathbf{0.68} \pm 0.012$ | $1.16 \pm 0.38$ | $1.27 \pm 0.092$ |
| Bike | 11,122 | 17 | $2.68 \pm 0.15$ | | $\mathbf{0.69} \pm 0.015$ | $1.17 \pm 0.38$ | $1.28 \pm 0.093$ |
| Kin40k | 25,600 | 8 | $1.44 \pm 0.028$ | | $\mathbf{0.71} \pm 0.045$ | $1.62 \pm 0.96$ | $3.26 \pm 0.23$ |
| Protein | 29,267 | 9 | $2.92 \pm 0.2$ | | $\mathbf{0.8} \pm 0.17$ | $2.27 \pm 0.9$ | $3.31 \pm 0.27$ |
| KeggDir | 31,248 | 20 | $7.14 \pm 0.39$ | | $\mathbf{0.85} \pm 0.1$ | $2.2 \pm 1.09$ | $3.8 \pm 0.38$ |
| CTslice | 34,240 | 385 | $52.01 \pm 0.92$ | | $\mathbf{3.32} \pm 5.0$ | $2.16 \pm 0.99$ | $3.87 \pm 0.34$ |
| KEGGU | 40,708 | 27 | $7.46 \pm 0.54$ | | $6.32^* \pm 0.41^*$ | $\mathbf{2.22} \pm 1.05$ | $4.78 \pm 0.4$ |
| 3DRoad | 278,319 | 3 | $\mathbf{1.93} \pm 0.12$ | | $126.37^* \pm 20.92^*$ | $12.01 \pm 5.51$ | $34.09 \pm 3.19$ |
| Song | 329,820 | 90 | $31.87 \pm 3.79$ | | $33.79^* \pm 10.45^*$ | $\mathbf{7.89} \pm 3.12$ | $39.55 \pm 3.08$ |
| Buzz | 373,280 | 77 | $\mathbf{20.18} \pm 6.66$ | | $571.15^* \pm 66.34^*$ | $29.25 \pm 18.33$ | $46.35 \pm 2.93$ |
| HouseElec | 1,311,539 | 11 | $\mathbf{118.41} \pm 3.93$ | | $575.64^* \pm 6.94^*$ | — | $367.71 \pm 4.7$ |

Table 1: NLL (top), RMSE (middle), and run time in minutes (bottom) on regression datasets, using a single GPU (Tesla V100-SXM2-16GB for BT and BTE and Tesla V100-SXM2-32GB for the other methods). The asterisk indicates an estimate of the time from the reported training time on 8 GPUS, assuming linear speedup in number of GPUs and independent noise in training times per GPU. All columns except BT and BTE come from Wang et al. [25].

## 8    Discussion

We have proven that the binary tree kernel GP is scalable. Our empirical results suggest that it often outpredicts not just other scalable methods, but even the popular Matérn GP. If the results in this paper replicate in other domains, it could obviate wide usage of classic GP kernels like the Matérn kernel, as well as inducing point kernels. Sometimes, our kernel fails to capture patterns in the data; some functions' values simply do not covary this way. But other kernels we tested seemed to fail like that even more.

Our contributions to linear algebra and kernel design may significantly increase the size of data sets on which GPs can do state-of-the-art modelling.

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
