# A Correctness of Algorithm 2

In this section, we show that Algorithm 2 performs approximate inversion. And then we show how to modify the algorithm in the setting were all columns of $U$ are identical, for a factor of $q$ speedup.

We begin by writing the exact form of $(u')^{p=l}$ from the proof of Theorem 1, and the corresponding $c'^{(l)}$. Recall the Sherman–Morrison Formula:

$$(A + cuu^\top)^{-1} = A^{-1} - \frac{A^{-1}uu^\top A^{-1}}{c^{-1} + u^\top A^{-1}u}$$

At the time that we update $A$ with the rank one matrix $c^{(l)}u^{p=l}u^{p=l^\top}$, what does $A$ equal? Let $c$ and $u$ originate from the $i^{\text{th}}$ column of $C$ and $U$. So our update is $C_{:,i}^{(l)}U_{:,i}^{P_{:,i}=l}U_{:,i}^{P_{:,i}=l^\top}$.

Then $A \leftarrow A_{i+1} = I + \sum_{k=i+1}^{q} \sum_{\ell \in [r]} C_{:,k}^{(\ell)}U_{:,k}^{P_{:,k}=\ell}U_{:,k}^{P_{:,k}=\ell^\top}$, recalling $r$ is the largest integer in $P$. And likewise, all the relevant entries of $C'$ and $U'$ have been calculated for $A^{-1}$. So we have $A_{i+1}^{-1} = I + \sum_{k=i+1}^{q} \sum_{\ell \in [r]} C'^{(\ell)}_{:,k}U'^{P_{:,k}=\ell}_{:,k}U'^{P_{:,k}=\ell^\top}_{:,k}$.

The main computation we need to do is $A^{-1}u$. We'll say that $k_\ell \sqsubseteq i_l$ if set $l$ from partition $P_{:,i}$ is a superset of set $\ell$ from partition $P_{:,k}$. That is, the rows where $P_{:,k}$ takes the value $\ell$ are a subset of the rows where $P_{:,i}$ takes the value $l$. The key simplification we use is that if $k_\ell \not\sqsubseteq i_l$, then $U_{:,i}^{P_{:,i}=l^\top}U'^{P_{:,k}=\ell}_{:,k} = 0$. The rows at which those two vectors have nonzero elements are disjoint. This logic is explained in the proof of Theorem 1 without all the notation.

So we let

$$U'^{P_{:,i}=l}_{:,i} = A_{i+1}^{-1}U_{:,i}^{P_{:,i}=l} = \left( I + \sum_{k=i+1}^{q} \sum_{\ell \in [r]} C'^{(\ell)}_{:,k}U'^{P_{:,k}=\ell}_{:,k}U'^{P_{:,k}=\ell^\top}_{:,k} \right) U_{:,i}^{P_{:,i}=l} \tag{4}$$

$$= \left( I + \sum_{k=i+1}^{q} \sum_{\ell:k_\ell \sqsubseteq i_l} C'^{(\ell)}_{:,k}U'^{P_{:,k}=\ell}_{:,k}U'^{P_{:,k}=\ell^\top}_{:,k} \right) U_{:,i}^{P_{:,i}=l} \tag{5}$$

$$= U_{:,i}^{P_{:,i}=l} + \sum_{k=i+1}^{q} \sum_{\ell:k_\ell \sqsubseteq i_l} C'^{(\ell)}_{:,k}U'^{P_{:,k}=\ell}_{:,k}U'^{P_{:,k}=\ell^\top}_{:,k}U_{:,i}^{P_{:,k}=\ell} \tag{6}$$

Equation 6 follows because the terms in the dot product $U'^{P_{:,k}=\ell^\top}_{:,k}U_{:,i}^{P_{:,i}=l}$ are only nonzero when $P_{:,k} = \ell$ and $P_{:,i} = l$, but the latter is implied by $k_\ell \sqsubseteq i_l$, so this is equivalent to the elements where $P_{:,k} = \ell$. So $U'^{P_{:,k}=\ell^\top}_{:,k}U_{:,i}^{P_{:,i}=l} = U'^{P_{:,k}=\ell^\top}_{:,k}U_{:,i}^{P_{:,k}=\ell}$, which gives us Equation 6. Then, we let

$$C'^{(l)}_{:,i} = \frac{-1}{1/C^{(l)}_{:,i} + U'^{P_{:,i}=l^\top}_{:,i}U_{:,i}^{P_{:,i}=l}} \tag{7}$$

**Proposition 3** (Correctness of Algorithm 2). *In Algorithm 2, $U'$ and $C'$ take the values defined in Equations 6 and 7.*

*Proof.* We'll assume that for $k > i$, $C'_{:,k}$ and $U'_{:,k}$ have the right values, and we'll show that $C'_{:,i}$ and $U'_{:,i}$ get the right values. Starting with $U'_{:,q}$, the $\sum_{k=q+1}^{q}$ in Equation 6 is empty, so $U'_{:,q} = U_{:,q}$. In Algorithm 2, $U'$ is initialized to $U$, and the $q^{\text{th}}$ column is never updated.

Now we see that $C'_{:,i}$ is correct assuming $U'_{:,i}$ is. In Line 5, we ensure $z_l = C^{(l)}_{:,i}U'^{P_{:,i}=l^\top}_{:,i}U_{:,i}^{P_{:,i}=l}$. A dot product is the sum of elementwise multiplications, and one can inspect that each such multiplication gets added to the right slot in $z$. Then, Line 6 implements Equation 7, with numerator and denominator multiplied by $C^{(l)}_{:,i}$. (It stores the same value in multiple locations).

Now we turn to $U'_{:,i}$. $U'_{:,i}$ is updated in every preceding loop. It starts out initialized to $U_{:,i}$, which accounts for the first term in Equation 6. In the sum from $k = i + 1$ to $q$, each term is accounted for

in a separate loop of the algorithm. We check that each term gets added at some point. So consider the term $\sum_{\ell:k_\ell \sqsubseteq i_l} C'^{(\ell)}_{:,k} U'^{P_{:,k}=\ell}_{:,k} U'^{P_{:,k}=\ell}_{:,k}{}^\top U^{P_{:,i}=l}_{:,i}$.

This term gets added to $U_{:,i}$ when $i$ from Algorithm 2 equals $k$ from Equation 6, and when $k$ from Algorithm 2 equals $i$ from Equation 6. We are very sorry about this correspondence, but it would have to happen either here or above in the discussion of $C'$. Observe that in Line 10, $y_{p_j k}$ takes the value $(U')^{P_{:,i}=p_j}_{:,i}{}^\top U^{P_{:,i}=p_j}_{:,k}$. Then, in Line 11, $y_{p_j k}$ gets multiplied by $(U')^{P_{:,i}=p_j}_{:,i}$ and $C'^{(p_j)}_{:,i}$, and added to $U'_{:,k}$. Swapping the $i$'s and $k$'s, and letting $p_j$ from Algorithm 2 equal $\ell$ from Equation 6, Lines 10 and 11 add to $U'_{:,k}$ the terms in Equation 6.

Thus, doing induction from $i = q$ down to 1, Algorithm 2 assigns $C'_{:,i}$ and $U'_{:,i}$ the correct values. $\square$

Now, we modify Algorithm 2 for the setting where all the columns $U_{:,i}$ are the same, allowing a speedup of $O(q)$.

---

**Algorithm 4** Inverse and determinant of $I+$ SROS Linear Operator, in which all columns of $U$ are the same.

---

**Require:** $P \in [r]^{n \times q}, C \in \mathbb{R}^{n \times q}, u \in \mathbb{R}^n$
**Ensure:** $I + L(P, C', U') = (I + L(P, C, u(\mathbf{1}^q)^\top))^{-1}; x = \log|I + L(P, C, u(\mathbf{1}^q)^\top)|$

1: $x, C', U' \leftarrow 0, \mathbf{0} \in \mathbb{R}^{n \times q}, u(\mathbf{1}^q)^\top$
2: **for** $i \in (q, q-1, ..., 1)$ **do**              $\triangleright O(nq)$ time
3:      $p, c, u' \leftarrow P_{:,i}, C_{:,i}, U'_{:,i}$
4:      $z \leftarrow \mathbf{0} \in \mathbb{R}^r$          $\triangleright z_i$ will store $c^{(l)}((u')^{p=l})^\top u^{p=l}$, where $c_k = c^{(l)}$ if $p_k = l$
5:      **for** $j \in [n]$ **do** $z_{p_j} \leftarrow z_{p_j} + c_j u'_j u_j$          $\triangleright O(n)$ time
6:      **for** $j \in [n]$ **do** $C'_{ji} \leftarrow -c_j/(1 + z_{p_j})$          $\triangleright O(n)$ time
7:      **for** $i \in [r]$ **do** $x \leftarrow x + \log(1 + z_i)$          $\triangleright O(n)$ time
8:      **if** i > 0 **then**
9:          **if** False **then**      $\triangleright$ This block is the slow version. What follows below is equivalent.
10:              $y \leftarrow \mathbf{0} \in \mathbb{R}^{n \times i}$
11:              **for** $j, k \in [n] \times [i-1]$ **do** $y_{p_j k} \leftarrow y_{p_j k} + u'_j u_j$
12:              **for** $j, k \in [n] \times [i-1]$ **do** $U'_{jk} \leftarrow U'_{jk} + C'_{ji} u'_j y_{p_j k}$
13:          $y \leftarrow \mathbf{0} \in \mathbb{R}^n$          $\triangleright O(n)$ time
14:          $U'_{:,(i-1)} \leftarrow U'_{:,i}$          $\triangleright O(n)$ time
15:          **for** $j \in [n]$ **do** $y_{p_j} \leftarrow y_{p_j} + u'_j u_j$          $\triangleright O(n)$ time
16:          **for** $j \in [n]$ **do** $U'_{j(i-1)} \leftarrow U'_{j(i-1)} + C'_{ji} u'_j y_{p_j}$          $\triangleright O(n)$ time
     **return** $C', U', x$

---

**Proposition 4** ($O(nq)$ inversion). *Algorithm 4 performs approximate inversion in $O(nq)$ time.*

*Proof.* The fact that Algorithm 4 runs in $O(nq)$ time is easily verified. Up to Line 12, Algorithm 4 is the same as Algorithm 2, except $U$ has been replaced with $u(\mathbf{1}^q)^\top$. So all we have to show is that Lines 13-16 produce the same result that Lines 10-12 would have.

Observe that at the start of the algorithm, $U'_{:,k}$ and $U'_{:,k+1}$ are initialized to the same value. Observe that Lines 11 and 12 repeat the same computation $i-1$ times. So in Lines 11 and 12, $U'_{:,k}$ and $U'_{:,k+1}$ are updated by the same amount if $i > k+1$, and they aren't updated at all if $i \leq k$. So the difference between $U'_{:,k}$ and $U'_{:,k+1}$ comes from only the former being updated when $i = k+1$.

Therefore, Line 14 initializes $U'_{:,k}$ to $U'_{:,k+1}$. Then Lines 15 and 16 update $U'_{:,k}$ with the appropriate difference; they copy Lines 11 and 12, setting $k$ to $i-1$. $\square$

## B   Gradient of loss with respect to weights

In this section, we show how to calculate $\nabla_w$ NLL in $O(nq^2)$ time. Recall:

$$\text{NLL}(w) = \frac{1}{2}\left(y^\top (K_{XX}(w) + \lambda I)^{-1}y + \log|K_{XX}(w) + \lambda I| + n\log(2\pi)\right). \tag{8}$$

Differentiating gives:

$$\frac{\partial\,\text{NLL}}{\partial w_i} = \frac{1}{2}\left[-y^\top (K_{XX} + \lambda)^{-1}\frac{\partial K_{XX}}{\partial w_i}(K_{XX} + \lambda)^{-1}y + \text{Tr}\left(\frac{\partial K_{XX}}{\partial w_i}(K_{XX} + \lambda)^{-1}\right)\right] \tag{9}$$

We begin by evaluuating $\partial K_{XX}/\partial w_i$. Recall from Proposition 2 that $K_{XX} = L(P, C, U)$, where $C = \mathbf{1}^n w^\top$, and $U = \mathbf{1}^{n\times q}$. It follows easily from the definition of $L$ that the elements of $L(P,C,U)$ are linear in the elements of $C$. So,

$$\frac{\partial K_{XX}}{\partial w_i} = L(P_{:,i}, \mathbf{1}^n, \mathbf{1}^n) \tag{10}$$

Algorithm 3 shows how to calculate $(K_{XX}+\lambda)^{-1}$ in $O(nq)$ time, and represent it as $L(P, C^{-1}, U^{-1})$. (Recall $C^{-1}$ and $U^{-1}$ are not true inverses; we just the notation to denote their purpose.) Now we turn to the question of how to calculate $\text{Tr}[L(P_{:,i}, \mathbf{1}^n, \mathbf{1}^n)L(P, C^{-1}, U^{-1})]$. We are considering symmetric matrices, so the trace of the product is the sum of the elements of the elementwise product. We expand and simplify:

$$T := \text{Tr}\left[L(P_{:,i}, \mathbf{1}^n, \mathbf{1}^n)L(P, C^{-1}, U^{-1})\right] \tag{11}$$

$$= \sum_{j=1}^{q}\text{Tr}\left[L(P_{:,i}, \mathbf{1}^n, \mathbf{1}^n)L(P_{:,j}, C_{:,j}^{-1}, U_{:,j}^{-1})\right] \tag{12}$$

$$\overset{(a)}{=} \sum_{j=1}^{q}\text{Tr}\left[L(P_{:,\max(i,j)}, \mathbf{1}^n, \mathbf{1}^n)L(P_{:,\max(i,j)}, C_{:,j}^{-1}, U_{:,j}^{-1})\right] \tag{13}$$

$$\overset{(b)}{=} \sum_{j=1}^{q}\sum_{\text{elements}} L(P_{:,\max(i,j)}, C_{:,j}^{-1}, U_{:,j}^{-1}) \tag{14}$$

$$\overset{(c)}{=} \sum_{j=1}^{q}\sum_{\ell=1}^{\max(P_{:,\max(i,j)})}\sum_{\text{elements}} (C_{:,j}^{-1})^{(\ell)}(U_{:,j}^{-1})^{P_{:,\max(i,j)}=\ell}(U_{:,j}^{-1})^{P_{:,\max(i,j)}=\ell\top} \tag{15}$$

$$= \sum_{j=1}^{q}\sum_{\ell=1}^{\max(P_{:,\max(i,j)})} (C_{:,j}^{-1})^{(\ell)}\|(U_{:,j}^{-1})^{P_{:,\max(i,j)}=\ell}\|_1^2 \tag{16}$$

where $(a)$ follows from the fact that the $(i,j)^{\text{th}}$ element of $L(p,c,u)$ is zero unless $p_i = p_j$, in which case, it is $c_i u_i u_j$; when multiplying elementwise by $L(p', c', u')$, where $p'$ is a finer partition, $L(p, c, u) \odot L(p', c', u') = L(p', c, u) \odot L(p', c', u')$, because $L(p, c, u)$ and $L(p', c, u)$ only differ on elements where $L(p', c', u')$ is 0 anyway. In the context of $(a)$, $P_{:,\max(i,j)}$ is a finer partition than $P_{:,\min(i,j)}$. $(b)$ follows because we are doing elementwise multiplication between the two matrices; anywhere $L(P_{:,\max(i,j)}, \mathbf{1}^n, \mathbf{1}^n)$ is 0, $L(P_{:,\max(i,j)}, C_{:,j}^{-1}, U_{:,j}^{-1})$ is already 0, and elsewhere, multiplying elements by 1 does not effect the matrix. $(c)$ follows from the construction of $L$.

It is straightforward to compute this in $O(nq)$ time. See Algorithm 5, which runs in $O(n)$ time and can be iterated over the $q$ terms in Equation 16.

Now we can see that Equation 9 can be computed for all $w_i$ in $O(nq^2)$ time. First, $(K_{XX} + \lambda)^{-1}$ can be computed in $O(nq)$ time, in the form $L(P, C^{-1}, U^{-1})$, as shown in Algorithm 3. Then, $z = (K_{XX} + \lambda)^{-1}y$ can be computed in $O(nq)$ time, also as shown in Algorithm 3. Then, for each $i \in [q]$, we can calculate $-z^\top \frac{\partial K_{XX}}{\partial w_i}z = -z^\top L(P_{:,i}, \mathbf{1}^n, \mathbf{1}^n)z$ in $O(n)$ time. Thus, handling the first term in Equation 9 for all $i$ takes a total of $O(nq)$ time. As just shown, the second term can be computed in $O(nq)$ time for each $i$, giving a total run time of $O(nq^2)$.

**Algorithm 5** Calculate $\sum_{\ell=1}^{\max(p)} c^{(\ell)} ||u^{p=\ell}||_1^2$.

---

**Require:** $p \in [r]^n, c, u \in \mathbb{R}^n$
**Ensure:** $x = \sum_{\ell=1}^m c^{(\ell)} ||u^{p=\ell}||_1^2$
  1: $y \leftarrow \mathbf{0}^m$
  2: **for** $j \in [n]$ **do** $y_{p_j} \leftarrow y_{p_j} + \sqrt{c_j} u_j$     ▷ $\sqrt{c_j}$ may be imaginary, but it will later be squared.
                                               With modifications, we could avoid complex types.
  3: $x \leftarrow 0$
  4: **for** $j \in [r]$ **do** $x \leftarrow x + y_j^2$
     **return** $x$

---

## C   Experimental Details

We initialize 160 random bit orders. For each one, we initialize three weight vectors $w$: uniform, uniform except the last bit is 0.5, and uniform except the last bit is 0.9. Out of these 480 initializations, we draw 20 samples via Boltzman sampling [5] on the log likelihood of the training data (after standardizing the values to have zero mean and unit variance). Then, we optimize the weights and bit order with BFGS as described in Section 5, using line search with Wofle conditions, with no extra gradient computations during line search. This allows fewer calculations of the gradient relative to the cheaper calculation of the loss. The BT column in Table 1 refers to the performance of the binary tree kernel, using the weights and bit order that gave the lowest training NLL out of these 20 trained models.

BTE produces a Gaussian mixture model at each predictive location, mixing over the predictive Gaussians produced by each of these 20 trained models. The relative weights of each Gaussian in the mixture depends on the training NLL of the model that produced it. We weight the models according to the softmax of the *per-data-point* NLL with a temperature of 0.01.

We follow the same train/test/validation splits as Wang et al. [25], but we never use the validation set, which the methods we compare against need. Thus, we could add the validation data to the training data for the binary tree kernel and call it a fair comparison, but we didn't do this, so as not to confuse the origin of the binary tree kernel's success.

## D   Additional Empirical Evaluation

### D.1   Sensitivity Analysis on Precision $p$

In Table 2, we evaluate the performance of the BT and BTE kernels on the precision $p$ for $p \in (2, 4, 8)$. In all problems, we find that RMSE and NLL decrease monotonically as $p$ increases and wall time increases monotonically. For best predictive performance, $p$ should be set as large as possible subject resource constraints. This validates that our heuristic rule for setting $p$ is a reasonable choice in a variety of settings.

### D.2   A Simple Performance Improvement

As shown in Table 3, the heuristic rule for setting $p$ results in bit strings that preserve the uniqueness of the raw training set for most datasets. However, for the 3dRoad, Song, and Buzz datasets, the percentage of unique bit strings is very low relative to the percentage of unique training rows. We note that this observation is from exploratory data analysis (EDA) and can be made before model fitting.

There are two key ways that mapping training rows to bit strings can lower the percent of unique examples. First, a low percentage of unique bit strings can arise if a given input feature has a non-uniform input distribution which can lead to multiple different inputs mapping to the same discrete bucket. To alleviate, this problem EDA can be used to determine a suitable feature transformation. For example, we apply a strictly increasing, piecewise linear transformation to the data, mapping the $k^{\text{th}}$ percentile of each dimension to $k/100$, for $k \in \{0, 10, 20, ..., 100\}$. This resembles an empirical cumulative density function (ECDF). Second, if the precision $p$ is set too low, then multiple different inputs can map to the same bit string. A simple solution is to iteratively increase the precision $p$

| DATASET | $n$ | $d$ | BTE ($p=2$) | BTE ($p=4$) | BTE ($p=8$) | BT ($p=2$) | BT ($p=4$) | BT ($p=8$) |
|---|---|---|---|---|---|---|---|---|
| POLETELE | 9,600 | 26 | $0.772 \pm 0.022$ | $-0.266 \pm 0.023$ | $-0.664 \pm 0.052$ | $0.771 \pm 0.021$ | $-0.198 \pm 0.012$ | $-0.398 \pm 0.159$ |
| ELEVATORS | 10,623 | 18 | $1.095 \pm 0.046$ | $0.761 \pm 0.024$ | $0.654 \pm 0.023$ | $1.097 \pm 0.049$ | $0.756 \pm 0.019$ | $0.662 \pm 0.018$ |
| BIKE | 11,122 | 17 | $1.095 \pm 0.046$ | $0.761 \pm 0.024$ | $0.654 \pm 0.023$ | $0.583 \pm 0.013$ | $-0.004 \pm 0.020$ | $-0.800 \pm 0.271$ |
| KIN40K | 25,600 | 8 | $0.888 \pm 0.004$ | $0.871 \pm 0.009$ | $0.869 \pm 0.004$ | $0.894 \pm 0.005$ | $0.879 \pm 0.012$ | $0.882 \pm 0.006$ |
| PROTEIN | 29,267 | 9 | $1.280 \pm 0.007$ | $1.042 \pm 0.012$ | $0.781 \pm 0.022$ | $1.281 \pm 0.007$ | $1.048 \pm 0.013$ | $0.842 \pm 0.032$ |
| KEGGDIR | 31,248 | 20 | $0.917 \pm 0.028$ | $-0.607 \pm 0.019$ | $-1.030 \pm 0.019$ | $0.916 \pm 0.029$ | $-0.608 \pm 0.017$ | $-1.028 \pm 0.023$ |
| CTSLICE | 34,240 | 385 | — | — | — | — | — | — |
| KEGGU | 40,708 | 27 | $0.228 \pm 0.057$ | $-0.607 \pm 0.009$ | $-0.668 \pm 0.007$ | $0.228 \pm 0.058$ | $-0.606 \pm 0.008$ | $-0.677 \pm 0.015$ |
| 3DROAD | 278,319 | 3 | $1.292 \pm 0.007$ | $0.973 \pm 0.007$ | $-0.255 \pm 0.004$ | $1.295 \pm 0.003$ | $0.981 \pm 0.004$ | $-0.251 \pm 0.005$ |
| SONG | 329,820 | 90 | $1.328 \pm 0.001$ | — | — | $1.317 \pm 0.014$ | — | — |
| BUZZ | 373,280 | 77 | $1.198 \pm 0.003$ | $1.107 \pm 0.009$ | — | $1.198 \pm 0.003$ | $1.106 \pm 0.009$ | — |
| HOUSEELEC | 1,311,539 | 11 | $0.629 \pm 0.003$ | $-0.673 \pm 0.003$ | $-2.569 \pm 0.006$ | $0.629 \pm 0.003$ | $-0.669 \pm 0.007$ | $-2.492 \pm 0.012$ |
| POLETELE | 9,600 | 26 | $0.513 \pm 0.012$ | $0.185 \pm 0.006$ | $0.159 \pm 0.004$ | $0.514 \pm 0.012$ | $0.194 \pm 0.003$ | $0.160 \pm 0.009$ |
| ELEVATORS | 10,623 | 18 | $0.725 \pm 0.031$ | $0.525 \pm 0.014$ | $0.481 \pm 0.016$ | $0.725 \pm 0.032$ | $0.520 \pm 0.013$ | $0.483 \pm 0.015$ |
| BIKE | 11,122 | 17 | $0.430 \pm 0.005$ | $0.237 \pm 0.005$ | $0.120 \pm 0.057$ | $0.431 \pm 0.006$ | $0.237 \pm 0.005$ | $0.104 \pm 0.029$ |
| KIN40K | 25,600 | 8 | $0.590 \pm 0.003$ | $0.582 \pm 0.005$ | $0.580 \pm 0.003$ | $0.593 \pm 0.004$ | $0.586 \pm 0.006$ | $0.587 \pm 0.005$ |
| PROTEIN | 29,267 | 9 | $0.870 \pm 0.006$ | $0.687 \pm 0.010$ | $0.609 \pm 0.008$ | $0.870 \pm 0.006$ | $0.691 \pm 0.010$ | $0.623 \pm 0.010$ |
| KEGGDIR | 31,248 | 20 | $0.604 \pm 0.017$ | $0.128 \pm 0.003$ | $0.086 \pm 0.003$ | $0.604 \pm 0.017$ | $0.128 \pm 0.003$ | $0.087 \pm 0.003$ |
| CTSLICE | 34,240 | 385 | — | — | — | — | — | — |
| KEGGU | 40,708 | 27 | $0.302 \pm 0.018$ | $0.128 \pm 0.002$ | $0.120 \pm 0.002$ | $0.302 \pm 0.018$ | $0.129 \pm 0.001$ | $0.119 \pm 0.002$ |
| 3DROAD | 278,319 | 3 | $0.882 \pm 0.005$ | $0.642 \pm 0.004$ | $0.187 \pm 0.000$ | $0.883 \pm 0.003$ | $0.645 \pm 0.002$ | $0.186 \pm 0.001$ |
| SONG | 329,820 | 90 | $0.914 \pm 0.001$ | — | — | $0.904 \pm 0.012$ | — | — |
| BUZZ | 373,280 | 77 | $0.801 \pm 0.002$ | $0.729 \pm 0.007$ | — | $0.801 \pm 0.002$ | $0.730 \pm 0.007$ | — |
| HOUSEELEC | 1,311,539 | 11 | $0.453 \pm 0.001$ | $0.121 \pm 0.001$ | $0.029 \pm 0.001$ | $0.453 \pm 0.001$ | $0.120 \pm 0.001$ | $0.029 \pm 0.001$ |
| POLETELE | 9,600 | 26 | $0.600 \pm 0.000$ | $2.500 \pm 0.200$ | $8.600 \pm 0.600$ | $0.600 \pm 0.000$ | $2.500 \pm 0.200$ | $8.600 \pm 0.600$ |
| ELEVATORS | 10,623 | 18 | $0.500 \pm 0.100$ | $1.600 \pm 0.100$ | $3.000 \pm 0.400$ | $0.500 \pm 0.100$ | $1.600 \pm 0.100$ | $3.000 \pm 0.400$ |
| BIKE | 11,122 | 17 | $0.400 \pm 0.000$ | $1.700 \pm 0.100$ | $3.100 \pm 0.200$ | $0.400 \pm 0.000$ | $1.700 \pm 0.100$ | $3.100 \pm 0.200$ |
| KIN40K | 25,600 | 8 | $0.300 \pm 0.000$ | $1.200 \pm 0.200$ | $1.500 \pm 0.000$ | $0.300 \pm 0.000$ | $1.200 \pm 0.200$ | $1.500 \pm 0.000$ |
| PROTEIN | 29,267 | 9 | $0.200 \pm 0.000$ | $0.600 \pm 0.000$ | $2.800 \pm 0.100$ | $0.200 \pm 0.000$ | $0.600 \pm 0.000$ | $2.800 \pm 0.100$ |
| KEGGDIR | 31,248 | 20 | $0.400 \pm 0.000$ | $2.300 \pm 0.100$ | $7.800 \pm 0.600$ | $0.400 \pm 0.000$ | $2.300 \pm 0.100$ | $7.800 \pm 0.600$ |
| CTSLICE | 34,240 | 385 | — | — | — | — | — | — |
| KEGGU | 40,708 | 27 | $0.800 \pm 0.100$ | $5.900 \pm 0.800$ | $12.400 \pm 1.200$ | $0.800 \pm 0.100$ | $5.900 \pm 0.800$ | $12.400 \pm 1.200$ |
| 3DROAD | 278,319 | 3 | $0.100 \pm 0.000$ | $0.200 \pm 0.000$ | $2.100 \pm 0.100$ | $0.100 \pm 0.000$ | $0.200 \pm 0.000$ | $2.100 \pm 0.100$ |
| SONG | 329,820 | 90 | $11.6 \pm 0.8$ | — | — | $11.6 \pm 0.8$ | — | — |
| BUZZ | 373,280 | 77 | $20.200 \pm 6.600$ | $54.900 \pm 13.600$ | — | $20.200 \pm 6.600$ | $54.900 \pm 13.600$ | — |
| HOUSEELEC | 1,311,539 | 11 | $3.300 \pm 0.200$ | $37.800 \pm 2.300$ | $118.41 \pm 3.93$ | $3.300 \pm 0.200$ | $37.800 \pm 2.300$ | $118.41 \pm 3.93$ |

Table 2: A sensitivity analysis of the performance of the BT kernel with respect to the precision $p$. NLL (top), RMSE (middle), and run time in minutes (bottom) on regression datasets, using a single GPU (Tesla V100-SXM2-16GB for BT and BTE and Tesla V100-SXM2-32GB for the other methods). Omitted results were not run due to limited GPU memory.

based on the difference between the percentage of unique rows in the raw training set and unique bit strings under precision $p$. The last column of Table 3 reports the percentage of unique bit strings on the 3dRoad, Song, and Buzz datasets after applying transforming each feature through its ECDF and increasing the precision if need be. These changes (all made through EDA) lead to significantly more unique bit strings. Table 4 shows the performance on these datasets under the proposed ECDF transformations and settings of $p$. We find that the BTE outperforms all methods on 3dRoad and Buzz with respect to RSME and NLL under these proposed changes. The performance on Song also improves.

| DATASET | $n$ | $d$ | $p$ | % UNIQUE TRAINING ROWS | % UNIQUE BIT STRINGS UNDER $p$ | ... AFTER TRANSFORM & W/ $p_{\text{NEW}}$ | $p_{\text{NEW}}$ |
|---|---|---|---|---|---|---|---|
| POLETELE | 9,600 | 26 | 6 | 99.9 | 98.8 | | |
| ELEVATORS | 10,623 | 18 | 8 | 100.0 | 100.0 | | |
| BIKE | 11,122 | 17 | 8 | 100.0 | 100.0 | | |
| KIN40K | 25,600 | 8 | 8 | 100.0 | 100.0 | | |
| PROTEIN | 29,267 | 9 | 8 | 97.4 | 96.5 | | |
| KEGGDIR | 31,248 | 20 | 8 | 36.1 | 36.1 | | |
| CTSLICE | 34,240 | 385 | 1 | 99.9 | 95.5 | | |
| KEGGU | 40,708 | 27 | 6 | 32.5 | 32.3 | | |
| 3DROAD | 278,319 | 3 | 8 | 99.3 | 15.9 | 97.7 | 16 |
| SONG | 329,820 | 90 | 2 | 100.0 | 46.2 | 100 | 2 |
| BUZZ | 373,280 | 77 | 2 | 98.5 | 0.5 | 94.6 | 3 |
| HOUSEELEC | 1,311,539 | 11 | 8 | 100.0 | 100.0 | | |

Table 3: Percentage of unique training inputs (over all training inputs) and bit strings (over all training inputs) under the precision $p$ set according to the heuristic rule. We reported the means across 3 training, validation, test set partitions. The last column shows percentage of unique bit strings after transforming each feature through its ECDF.

| Dataset | $n$ | $d$ | BTE | BT | Matérn (BBMM) | SGPR | SVGP |
|---|---|---|---|---|---|---|---|
| 3DRoad | 278,319 | 3 | $\mathbf{-1.285} \pm 0.008$ | $-1.267 \pm 0.005$ | $0.909 \pm 0.001$ | $0.943 \pm 0.002$ | $0.697 \pm 0.002$ |
| Song | 329,820 | 90 | $1.306 \pm 0.011$ | $1.331 \pm 0.003$ | $\mathbf{1.206} \pm 0.024$ | $1.213 \pm 0.003$ | $1.417 \pm 0.000$ |
| Buzz | 373,280 | 77 | $\mathbf{0.017} \pm 0.002$ | $0.034 \pm 0.000$ | $0.267 \pm 0.028$ | $0.106 \pm 0.008$ | $0.224 \pm 0.050$ |
| 3DRoad | 278,319 | 3 | $\mathbf{0.104} \pm 0.002$ | $\mathbf{0.105} \pm 0.002$ | $\mathbf{0.101} \pm 0.007$ | $0.661 \pm 0.010$ | $0.481 \pm 0.002$ |
| Song | 329,820 | 90 | $0.894 \pm 0.010$ | $0.904 \pm 0.012$ | $\mathbf{0.807} \pm 0.024$ | $\mathbf{0.803} \pm 0.002$ | $0.998 \pm 0.000$ |
| Buzz | 373,280 | 77 | $\mathbf{0.249} \pm 0.001$ | $0.253 \pm 0.000$ | $0.288 \pm 0.018$ | $0.300 \pm 0.004$ | $0.304 \pm 0.012$ |
| 3DRoad | 278,319 | 3 | $14.2 \pm 0.2$ | | $126.37^* \pm 20.92^*$ | $12.01 \pm 5.51$ | $34.09 \pm 3.19$ |
| Song | 329,820 | 90 | $11.5 \pm 0.7$ | | $33.79^* \pm 10.45^*$ | $\mathbf{7.89} \pm 3.12$ | $39.55 \pm 3.08$ |
| Buzz | 373,280 | 77 | $82.8 \pm 5.7$ | | $571.15^* \pm 66.34^*$ | $\mathbf{29.25} \pm 18.33$ | $46.35 \pm 2.93$ |

Table 4: NLL (top), RMSE (middle), and run time in minutes (bottom) on regression datasets, using a single GPU (Tesla V100-SXM2-16GB for BT and BTE and Tesla V100-SXM2-32GB for the other methods) *after transforming each feature through its ECDF and using precision $p_{new}$*. The asterisk indicates an estimate of the time from the reported training time on 8 GPUS, assuming linear speedup in number of GPUs and independent noise in training times per GPU.

## D.3 Comparison with Simplex-GP

Here, we compare against Simplex-GP [15] using the 4 datasets that are common to both papers. We find that BTE outperforms Simplex GP on all datasets with respect to NLL and on 3 out of 4 datasets with respect to RMSE. Furthermore, Simplex-GP is not the top performing method on any dataset.

| Dataset | $n$ | $d$ | BTE | BT | Matérn (BBMM) | SGPR | SVGP | Simplex-GP |
|---|---|---|---|---|---|---|---|---|
| Elevators | 10,623 | 18 | $0.649 \pm 0.032$ | $0.646 \pm 0.023$ | $0.619 \pm 0.054$ | $0.580 \pm 0.060$ | $\mathbf{0.519} \pm 0.022$ | $1.600 \pm 0.020$ |
| Protein | 29,267 | 9 | $\mathbf{0.781} \pm 0.023$ | $0.845 \pm 0.026$ | $1.018 \pm 0.056$ | $0.970 \pm 0.010$ | $1.035 \pm 0.006$ | $1.406 \pm 0.048$ |
| KeggDir | 31,248 | 20 | $-1.031 \pm 0.020$ | $-1.029 \pm 0.021$ | $-0.199 \pm 0.381$ | $\mathbf{-1.123} \pm 0.016$ | $-0.940 \pm 0.020$ | $0.797 \pm 0.031$ |
| HouseElec | 1,311,539 | 11 | $\mathbf{-2.569} \pm 0.006$ | $-2.492 \pm 0.012$ | $-0.152 \pm 0.001$ | — | $-1.010 \pm 0.039$ | $0.756 \pm 0.075$ |
| Elevators | 10,623 | 18 | $0.478 \pm 0.021$ | $0.476 \pm 0.018$ | $\mathbf{0.394} \pm 0.006$ | $0.437 \pm 0.018$ | $\mathbf{0.399} \pm 0.009$ | $0.510 \pm 0.018$ |
| Protein | 29,267 | 9 | $0.608 \pm 0.008$ | $0.623 \pm 0.011$ | $\mathbf{0.536} \pm 0.012$ | $0.656 \pm 0.010$ | $0.668 \pm 0.005$ | $0.571 \pm 0.003$ |
| KeggDir | 31,248 | 20 | $\mathbf{0.086} \pm 0.003$ | $\mathbf{0.086} \pm 0.003$ | $\mathbf{0.086} \pm 0.005$ | $0.104 \pm 0.003$ | $0.096 \pm 0.001$ | $0.095 \pm 0.002$ |
| HouseElec | 1,311,539 | 11 | $\mathbf{0.029} \pm 0.001$ | $\mathbf{0.029} \pm 0.001$ | $0.055 \pm 0.000$ | — | $0.084 \pm 0.005$ | $0.079 \pm 0.002$ |

Table 5: NLL (top), RMSE (middle), and run time in minutes (bottom) on regression datasets, using a single GPU (Tesla V100-SXM2-16GB for BT and BTE and Tesla V100-SXM2-32GB for the other methods). The asterisk indicates an estimate of the time from the reported training time on 8 GPUS, assuming linear speedup in number of GPUs and independent noise in training times per GPU. All columns except BT and BTE come from Wang et al. [25] and Simplex-GP results come from Kapoor et al. [15].

## E Societal Impacts

More accurate machine learning models allow for better decision making. Improvements in decision making can lead to improved societal outcomes in many applications such as early disease detection. However when predictions of a machine learning algorithm are confident, we may be more compelled to act on them, which could lead to increased risk in delicate settings. Therefore, confident and wrong predictions, resulting from poor and confident generalization due to overfitting, can lead to worse outcomes. When using binary tree kernels for Gaussian processes, overfitting is possible. That said, Gaussian processes do quantify their uncertainty very naturally, which can be helpful for knowing when to take their predictions with a grain, or a heap, of salt. As with all machine learning models, what outcomes are predicted and how those predictions are used are ultimately decided by to the practitioner, and those decisions contribute to whether the model impacts society in positive or negative ways.