# OpenReview forum: "Log-Linear-Time Gaussian Processes Using Binary Tree Kernels"
_NeurIPS.cc/2022/Conference — NeurIPS 2022 Accept_

### Official Review · Reviewer_9UL1 · 2022-07-07

**Rating:** 6
**Confidence:** 4
**Soundness:** 2 fair
**Presentation:** 1 poor
**Contribution:** 3 good

**Summary:**

The paper discusses a new type of kernel for Gaussian process regression based on binary-tree partitions of the input space. The structure of the kernel allows for a fast (linear time) inversion of the kernel matrix which is benefitable for large datasets. The Gaussian process with this kernel gives piecewise-constant predictions.

**Questions:**

Since the authors claim the superioirity of their kernel over other methods of scaling up GP regression, it is interesting how the experiments were carried out.
In specific, it is known that a Matern kernel performance depends on the choice of the smoothness parameter. How did you choose this parameter in your experiments?

It is also interesting how did you choose the $\lambda$ parameter of your kernel (which is basically noise level)? Is it possible to train this parameter along with the order and weights?

Also an interesting thing is how does the choice of the precision $p$ affect the performance? In particular, since the train time scales quadratically with the precision. How did you come up with the precision in the experiments and is it possible to automatically train the precision? Does it makes sense to ensemble over not only different bit orders but also over different precisions?

It would be nice to see some comparision ageints other partition-based methods like Random Forest Kernel.


**Limitations:**

Authors have clearly stated the limitations of their work. The predictions provided with a GP with binary-tree kernel are piecewise-constant. That makes the kernel unusable in Bayesian Optimization. It would be nice to discuss limitations of the choise of the precision level $p$ and noise level $\lambda$.

**Strengths And Weaknesses:**

*Originality*. The work is original. It is somewhat similar in nature to works related to Random Forest kernels but I am not aware of other work based on bit representations of the input data.

*Clarity*. One area where the paper could use improvement is the clarity, especially notation-wise. It is quite hard to read the heavy-weight superscripts throughout the proof of the theorem 1 in particular. It seems like there is a typo on the line 178: missing \prime? There is also some notation inconsistencies like using the same letters $i$, $j$, $k$ with a different meaning in similar contexts. Also it seems like the proof of the Theorem 1 in the main paper is in between a high-level sketch and a precise mathematical formulation which makes it harder to read. Could be benefitial if the proof outlined a high-level sketch with the details hidden in the supplement.

*Quality*. The quality of research is high. The authors provided proofs of the mathematical statements they make. The authors have also made empirical performance comparison against popular GP kernels, such as Matern or SVGP method. The authors have provided a review of a related and competing methods of scaling up GP regression. An interesting thing would be to compare their method to other partition-based methods, such as a Random Forest Kernel.

The authors claim that their kernel widely outperforms other methods such as exact regression with a Matern kernel or a sparse GP regression. However, as presented in the Table 1, this is perhaps an overstatement. In particular, Matern kernel beats the BT(E) for moderately-sized datasets both in test RMSE and computational speed, and in terms of the test NLL, it is not a clear cut that a BT kernel is ultimately a better choice.

It would be also interesting to see some ablation studies regarding the choice of the bit precision and the noise variance $\lambda$.

*Significance*. The work provides a new type of kernel that could be used in setttings where a piecewise-constant predictions are suitable. The authors provide an algorithm for a computationally-easy inversion of the (noised) covariance matrix. Since GP regression is notoriously heavy on the computational side, the new method to speed it up is a significant contribution.

---

> ### Author Response · Authors · 2022-08-02
> **Response to concerns (part 1)**
>
> $>$ One area where the paper could use improvement is the clarity, especially notation-wise. It is quite hard to read the heavy-weight superscripts throughout the proof of the theorem 1 in particular.  There is also some notation inconsistencies like using the same letters $i$, $j$, $k$ with a different meaning in similar contexts.
>
> Another reviewer agrees we should not to reuse the same letters in different settings. We have fixed this. $i$ now always indexes over $[q]$, $j$ now always indexes over $[n]$, and we have added a new symbol $l \in [r]$ for indexing over parts of a partition, with $r$ replacing one of the usages of $m$. For notation like $u^{p=j}$ (now $u^{p=l}$), we're not sure how reduce the size of the superscripts. $u^p_l$ would be confusing because then it looks like it should be the $l$\textsuperscript{th} element of some vector. $F(u, p, l)$ would be valid, but to our eyes, less intuitive.
>
> $>$ It seems like there is a typo on the line 178: missing ${}\prime$?
>
> Yes, thank you!
>
> $>$ Also it seems like the proof of the Theorem 1 in the main paper is in between a high-level sketch and a precise mathematical formulation which makes it harder to read. Could be benefitial if the proof outlined a high-level sketch with the details hidden in the supplement.
>
> We could convert phrases like "$(u')^{p=j}$ has the same sparsity pattern as $u^{p=j}$" into "$ (u')^{p=j}_i = 0 \iff (u')^{p=j}_i = 0$", but we think the former statement is unambiguous. Maybe more tentatively, we also feel that sentences like the following are unambiguous: "the extra component of $A_i^{-1}$ that appears after the $u^{p=j'}$
> update is irrelevant to the $u^{p=j}$ update." Especially if we added the clause "because $(u^{p=j})^{\top} (u')^{p=j'} = 0$". If you feel strongly about this, or if other reviewers agree, we'll do as you suggest.
>
> $>$ An interesting thing would be to compare their method to other partition-based methods, such as a Random Forest Kernel.
>
> We have added a citation to this method, pointing out that like ours, it incorporates binary trees. We agree it would be interesting; however, we feel there are too many kernels with this level of popularity to test against them all, and the formal connection to our method is limited.
> % Incidentally, our understanding from the paper is that the RF kernel is not actually a kernel. $k(x_1, x_2)$ is not well-defined; instead it depends on the other $x$ and $y$ values from the dataset.
>
> $>$ The authors claim that their kernel widely outperforms other methods such as exact regression with a Matern kernel or a sparse GP regression. However, as presented in the Table 1, this is perhaps an overstatement.
>
> It would be an overstatement to say that our kernel widely outperforms other methods. The strongest claim of ours we can find is: "But given that the binary tree kernel usually outperforms the Mat\'ern, we'll tentatively say the best first guess is that a new dataset has more binary tree character." This is in the context of a discussion of NLL, but maybe we should clarify that it usually outperforms the Mat\'ern \textit{with respect to NLL}. In the very next paragraph, we say that if one is trying to minimize RMSE, a Mat\'ern kernel might be better if it's tractable.
>
> $>$ It is also interesting how did you choose the $\lambda$ parameter of your kernel (which is basically noise level)? Is it possible to train this parameter along with the order and weights?
>
> We kind of did; if the last weight in the weight vector increases by $x$, and $\lambda$ decreases by $x$, the resulting kernel is the same. Since the last weight in the weight vector is optimized, it doesn't really matter what $\lambda$ is, except for the fact that the weight must be non-negative. So when we set $\lambda$, we are just setting the minimum noise level. We didn't set it to be a minimal value because we were concerned about overfitting when $n$ is small, but one could set it to a minimal value (constrained only by needing to avoid numerical errors in Algorithm 2) to fully optimize the noise level.
>
> $>$ It would be also interesting to see some ablation studies regarding the choice of the bit precision and the noise variance $\lambda$.
>
> We have added an analysis of how performance depends on choice of $p$, which we discuss in a comment to all reviewers. For the noise level, this is basically optimized as described above.

---

> > ### Author Response · Authors · 2022-08-02
> > **Reply to concerns (part 2)**
> >
> > $>$ In specific, it is known that a Matern kernel performance depends on the choice of the smoothness parameter. How did you choose this parameter in your experiments?
> >
> > Every column of Table 1 beside the BT and BTE columns are copied from Wang et al. [21]. We probably should have said this more clearly, sorry. (We said, "We evaluate our method on the same open-access UCI datasets [3] as Wang et al. [21], using
> > their same training, validation, and test partitions, and we compare against the baseline results they
> > report.") Since their contribution was to the Mat\'ern kernel (using BBMM), they probably trained the smoothness parameters well.
> >
> > $>$ Also an interesting thing is how does the choice of the precision  affect the performance? In particular, since the train time scales quadratically with the precision. How did you come up with the precision in the experiments and is it possible to automatically train the precision? Does it makes sense to ensemble over not only different bit orders but also over different precisions?
> >
> > We chose $p$ from a rough idea of how much time we wanted to spend on computation. There isn't any gradient-based way to train $p$, but it could absolutely be optimized. Our sensitivity analysis suggests larger is better, if there is time for it. A simple approach is to consider the percentage of unique bit strings as mentioned in the response to all reviewers. Such an approach makes it simple to increase the precision until computational limits are reached or the percentage of unique bit strings is satisfactorily large. All of this can be done without training a GP, simply by computing percentage of unique bit string under different precisions on the training set. It may also make sense to do an ensemble over different precisions.

---

> > > ### Comment · Reviewer_9UL1 · 2022-08-04
> > > **Reply to the authors**
> > >
> > > Thank you for your reply. A couple of points I want to make:
> > >
> > > >> It is also interesting how did you choose the parameter of your kernel (which is basically noise level)? Is it possible to train this parameter along with the order and weights?
> > > >
> > > > We kind of did; if the last weight in the weight vector increases by $x$, and $\lambda$ decreases by $x$, the resulting kernel is the same.
> > >
> > > I kindly disagree that the last weight serves as a noise level. The kernel matrix $K_{XX}+\lambda$ is the same for the training points but not the same for the test points (where $\lambda$ is not accounted for). If anything, the last weight serves as a means to quantify variance (or output scale) of the kernel, is it not?
> > >
> > > >> Also it seems like the proof of the Theorem 1 in the main paper is in between a high-level sketch and a precise mathematical formulation which makes it harder to read. Could be benefitial if the proof outlined a high-level sketch with the details hidden in the supplement.
> > > >
> > > > If you feel strongly about this, or if other reviewers agree, we'll do as you suggest.
> > >
> > > I don't. The other reviewers seem to be fine about this as well.
> > >
> > > All other points are acknowledged.

---

> > > > ### Author Response · Authors · 2022-08-09
> > > > **Reply**
> > > >
> > > > Ah this is very helpful; we had not thought about that. When writing the code, we shared your opinion that $\lambda$ should not be added to the predictive variance for the test set, and later, when writing the paper, we decided $\lambda$ should be added for the test points, but forgot that we had ever thought differently. See in Line 243 (or 242 of our recent revision) that the predictive covariance matrix $\Sigma_{X’}$ includes the term $\lambda I_m$, and see Line 17 of Algorithm 3. Thank you for helping us notice the discrepancy. Considering the matter again, we decided to change the code, rather than the paper, so it (now) is the case that the last weight indeed has a similar effect to $\lambda$. Our reasoning for adding $\lambda$ to the predictive variance is as follows: if the training data is produced by a noisy observation of a function, then so is the test data. To minimize nll, we are not trying to predict the function value at the test point; we are trying to predict the (noisy) observation there. The learned weights for the kernel are learned under the assumption that noise will be added to help model the training data, so the kernel will “expect” the same noise to help predict the test points. In some contexts, we would care more about our uncertainty around function values than our uncertainty about subsequent observations, so we’ve put both options into the code.
> > > >
> > > > Now for all that theory, having changed the code to add lambda to the predictive variance, the results are essentially same as they were. Again, thank you very much for raising this issue, as we hadn’t properly considered it.

---

### Official Review · Reviewer_a9uh · 2022-07-10

**Rating:** 6
**Confidence:** 3
**Soundness:** 2 fair
**Presentation:** 2 fair
**Contribution:** 3 good

**Summary:**


In this paper, the authors propose a binary tree-based kernel for Gaussian Process acceleration.  The proposed kernel enables a fast computation of the negative log-likelihood in $O(nq)$ time.

**Questions:**


1. Could the authors explain more about the time complexity regarding the above concern?

2. It is better to provide a detailed comparison with a strong inducing pionts-based GP (Kapoor et al. 2021) for better evaluation.


Kapoor et al. SKIing on Simplices: Kernel Interpolation on the Permutohedral Lattice for Scalable Gaussian Processes. ICML 2021.

**Strengths And Weaknesses:**


Pros.
1.  The proposed binary tree-based kernel enables a fast (approximate) compuation of  $(K_{X,X} + \lambda I)^{-1}$ and $|K_{X,X} + \lambda I |$, which is important for GP accelration.

2.  The fast computation algorithm (Algorithm 1) may be potential for acceleration in other tasks relying on the binary tree-based features.



Cons.
1.  The time complexity of Algorithm 2 may be up to $O(2^q)$.
 From Eq.(3),  the number of $s \in \mathbb{B}^r$ is $2^r$, and it can be up to $2^q$. As a result, the number of partitions can be as
large as $2^q$. The time complexity of the  for loop in step 7 in Algorithm 2  is then up to $O(2^q)$. This can be much larger than $O(nq)$.
Note that $q=dp$,  $O(2^q)$ is quite large.

2. The authors argue that inducing points-based GP acceleration is less competitive. However,  in the experiments,  a detailed comparison with strong inducing points-based methods is missing.   It is unconvinced to support the claim.

3. Some symbols are confusing.
(1). the index $i$ in $(X_{[s]})_i$ confuse with the index in $w_i$ in  Eq.(3).
(2). The $\boldsymbol{i}$ in step 8 in Algorithm 2 is not defined or used in Algorithm 2.
(3).  $m$ in Line 237-251 is confused with the number of partitions in Definition 2 in Line 120.

---

> ### Author Response · Authors · 2022-08-02
> **Reply to concerns**
>
> $>$ The time complexity of Algorithm 2 may be up to $O(2^q)$. From Eq.(3), the number of $s \in \mathbb{B}^r$ is $2^r$, and it can be up to $2^q$. As a result, the number of partitions can be as large as $2^q$. The time complexity of the for loop in step 7 in Algorithm 2 is then up to $O(2^q)$. This can be much larger than $O(nq)$. Note that $q=dp$,  $O(2^q)$ is quite large.
>
> We did not explain this well. The number of parts in a partition over binary strings is at most $2^q$--the number of binary strings of length $q$, but it is also at most $n$--the number of binary strings in the set we are partitioning. Many of those terms in Eq. (3) can be ignored. Thus, we say (confusingly) about step 7 in Algorithm 2 that it takes $O(n)$ time, even though it is a loop over $m$ elements. Here, $m$ is the maximum number of parts in a partition over $n$ binary strings of length $q$, which you were saying could be as high as $2^q$, but which we can bound by $n$.
>
> Note that $P \in [m]^{n \times q}$. The main reason we sort the data first is so that the numbers labelling each part of a given partition can be selected from $[n]$ rather than from $[2^q]$, the latter of which we could do without sorting.
>
> $>$ It is better to provide a detailed comparison with a strong inducing points-based GP (Kapoor et al. 2021) for better evaluation.
>
> Kapoor et al. report RMSE and NLL on four of the datasets that we use, using the same train/test splits, so we have compared their results to ours in Table 5 of Appendix D. On NLL, BTE always outperforms SIMPLEX-GP. On RMSE, BTE is better than SIMPLEX-GP on 3/4 datasets. But Simplex-GP is never the best method on any dataset with respect to NLL or RMSE, losing to the Mat\'ern kernel in the one setting where it beats ours. They don't report run time in a way that allows us to compare (they report maximum and minimum times for an epoch of training), so we omit this information.
>
> $>$ Some symbols are confusing...
>
> You are right that we use symbols like $i$ and $m$ differently in different contexts. Another reviewer agrees this is confusing. We have fixed this. $i$ now always indexes over $[q]$, $j$ now always indexes over $[n]$, and we have added a new symbol $l \in [r]$ for indexing over parts of a partition, with $r$ replacing one of the usages of $m$. One quick correction: you say "The $i$ in step 8 in Algorithm 2 is not defined or used", but it is defined in step 2 and used in many other lines. It looks like we used the wrong font though, writing i instead of $i$; sorry if that was the source of confusion.

---

> > ### Comment · Reviewer_a9uh · 2022-08-05
> > **Reply**
> >
> >
> > Thanks for the authors' detailed response.  My main concern has been addressed.  Overall,  I think the paper provides a novel scheme for GP acceleration. I am happy to raise my score.  However,  I still have some concerns.
> >
> > 1.   As the authors response,   the actual number of partitions is bounded by $n$, i.e., $r=min(n,2^q)$.  Does it mean that the kernel actually produces a $r=min(n,2^q)$-dimensional RKHS subspace instead of $2^q$ dimensional RKHS (stated in Line 271)?
> >
> > 2.  Simplex-GP is also a linear time GP acceleration method.  I am just curious whether the binary tree GP is faster than Simplex-GP when achieving a competitive NLL.

---

> > > ### Author Response · Authors · 2022-08-09
> > > **Reply**
> > >
> > > Thank you very much for raising your score, and thank you again for your time and thought and engagement.
> > >
> > > In the kernel literature, "infinite dimensional" basically means the kernel matrix is rank n no matter how large n is. Consider a Matern kernel for instance; an $n \times n$ kernel matrix cannot have a rank higher than n, but we still say the Matern kernel is infinite dimensional, rather $n$ dimensional. Formally, the dimensionality of an RKHS is the number of linearly independent functions of the form $k(\cdot, x)$ that we can find (using different values of $x$). This definition makes no reference to the number of data points $n$ that we happen to be concerned with. Alternatively, there should be some way to define the dimensionality of an RKHS as the limit of the rank of the kernel matrix as $n \to \infty$, although the details might be a bit messy. Either understanding of RKHS-dimensionality (number of linearly independent functions or upper bound on the rank of the kernel matrix as n grows) leads us to the conclusion that the RKHS is $2^q$-dimensional.
> > >
> > > > 	Simplex-GP is also a linear time GP acceleration method. I am just curious whether the binary tree GP is faster than Simplex-GP when achieving a competitive NLL.
> > >
> > > Yes, we do seem to outperform Simplex-GP on NLL, but the paper doesn’t exactly tell us the run times for Simplex-GP. They report a range of training times per epoch, and the number of epochs is capped at 100, and it looks like they do all 100 epochs of training. Taking the minimum epoch training time for all four datasets, and multiplying by 100, we get 750 minutes for House Electric, 67 for Kegg Directed, 13 for Protein, and 30 for Elevators. We may be misunderstanding the number of epochs actually required, and it may be half that, but even so, our method would be twice as fast or more.

---

### Official Review · Reviewer_GqxJ · 2022-07-11

**Rating:** 7
**Confidence:** 4
**Soundness:** 4 excellent
**Presentation:** 4 excellent
**Contribution:** 4 excellent

**Summary:**

The paper develops a novel binary tree kernel function for fast Gaussian process computation. The kernel is based on the similarity of permuted bit strings of the two input features, which induces a kernel matrix with a sparse rank one sum representation. The computational gain is achieved by efficient algorithms for matrix inversion and matrix-vector multiplication.

**Questions:**

Major comments:

1. As far as I understand, for any given bit permutation, $||x_1^{raw} - x_2^{raw}||_2 \rightarrow 0$ does not imply $k_w(x_1, x_2) \rightarrow ||w||_1$, where $x_1^{raw}, x_2^{raw} \in \mathbb{R}^d$ are the raw features, and $x_1, x_2 \in \mathbb{R}^q$ are their bit representation. However, in some low-dimensional problems such as spatial applications, it makes sense that data with two closer features (in terms of Euclidean distance) have stronger correlation, which cannot be captured by the proposed kernel. I am curious how the proposed method would perform in such scenarios.

2. It appears to me that after the bit permutation is updated during model training, we need to re-sort the bit feature matrix $X$ to obtain a new $P$. Does this mean the actual training time complexity is $O(n\log{n})$ instead of $O(n)$?

3. The proposed kernel seems to be highly nonstationary in nature, which could be one of the reasons why it outperforms the standard stationary Mat\'ern model in real applications where the stationarity assumption does not make sense. It would be interesting to see how it compares with other nonstationary GP models such as Zhang \& Williamson (2019) and Trapp et al. (2020).

4. I think the high-dimensional kernel parameter $\theta$ could be somewhat hard to learn. How sensitive is the performance to the initial values of $\theta$? Also, the performance of BT is based on multiple initializations, which may not be fair to other competing methods.


Minor comments:

* Could you please elaborate more on why the matrix inversion algorithm is called "approximate" inversion?
* Figure 2: Are those piecewise constant functions the posterior means of binary tree kernel GPs? I think a realization from the proposed GP should be piecewise smooth rather than piecewise constant.


References:

Trapp, M., Peharz, R., Pernkopf, F., \& Rasmussen, C. E. (2020). Deep structured mixtures of gaussian processes. In International Conference on Artificial Intelligence and Statistics (pp. 2251-2261). PMLR.

Zhang, M. M., \& Williamson, S. A. (2019). Embarrassingly parallel inference for Gaussian processes. Journal of Machine Learning Research.

**Limitations:**

Please see the major comments above.

**Strengths And Weaknesses:**

Strength:
* Instead of approximating a kernel matrix (or its inverse), the paper presents a novel idea of constructing a kernel with a computationally favorable structure.
* The proposed sparse rank one sum representation and algorithms could be applicable to other settings.
* The method is nicely illustrated.

Weakness (see major comments below for details):
* Unclear performance in low-dimensional settings.
* Training time complexity appears to be $O(n\log{n})$ instead of $O(n)$.
* The paper could benefit from additional experiment results.

---

> ### Author Response · Authors · 2022-08-02
> **Reply to concerns (part 1)**
>
> $>$ Unclear performance in low-dimensional settings.
>
> Out of the four datasets with the fewest dimensions, our method has the lowest NLL in three of them. But with respect to RMSE, your point stands (ours is only lowest on one of the four). However, as stated in the response to all reviewers, we show Appendix D (Table 4) that using a simple feature transformation and increasing the precision leads to improved performance on RMSE for 3dRoad, the lowest-dimensional dataset with 3 features. After that transformation, the BTE kernel performs the best of any method with respect to RMSE and NLL on this dataset.
>
> $>$ Training time complexity appears to be $O(n \log n)$ instead of $O(n)$.
>
> Yes, sorry. We had been focusing on "the GP part" of the problem, and taking sorted data for granted, but really you're right that the method as a whole is $O(n \log n)$, especially since, as you mention, we re-sort the data during training. We've updated the paper to reflect this. And we've changed the title to Log-linear.
>
> $>$ As far as I understand, for any given bit permutation,
> $||x^\textrm{raw}_1 - x^\textrm{raw}_2|| \to 0$ does not imply $k_w(x_1, x_2) \to ||w||_1$, where
> $x^\textrm{raw}_1, x^\textrm{raw}_2 \in \mathbb{R}^d$ are the raw features, and $x_1, x_2 \in \mathbb{R}^q$ [you must have meant $\mathbb{B}^q$?] are their bit representation. However, in some low-dimensional problems such as spatial applications, it makes sense that data with two closer features (in terms of Euclidean distance) have stronger correlation, which cannot be captured by the proposed kernel.
>
> It does in fact imply this almost everywhere. Let's say we have precision $p$. (Recall $q = pd$). As $x^\textrm{raw}_1$ and $x^\textrm{raw}_2$ approach each other, eventually the first $p$ bits of every coordinate must become equal, unless any of the coordinates of their meeting point are exactly $n/2^p$ for some integer $n$. But almost everywhere, as $x^\textrm{raw}_1$ and $x^\textrm{raw}_2$ approach, after some finite time, $x_1 = x_2$, which implies $k_w(x_1, x_2) = ||w||_1$. So this kernel can capture low-dimensional spatial applications. That said, if we know that function values are likely to covary as a function of Euclidean distance, then a Mat\'ern kernel would probably make more sense (if we have don't have too large a dataset).
>
> $>$ The proposed kernel seems to be highly nonstationary in nature, which could be one of the reasons why it outperforms the standard stationary Mat\'ern model in real applications where the stationarity assumption does not make sense. It would be interesting to see how it compares with other nonstationary GP models such as Zhang \& Williamson (2019) and Trapp et al. (2020).
>
> Running Zhang et al.'s ISMOE on PoleTele (the smallest dataset), using the same 3 random splits of the data, we find significantly worse NLL (ranging from 0.5 to 0.8 instead of $\sim$-0.2 for Mat\'ern and $\sim$-0.6 for BTE) and RMSE (ranging from 0.5 to 1 instead of $\sim$0.15 for Mat\'ern and BTE). On the next smallest dataset where we outperform the stationary kernels, Bike, ISMOE again performed significantly worse (NLL ranging from 0.54 to 1.36 instead of 0.1 for Mat\'ern and -0.8 for ours, and RMSE ranging from 0.6 to 0.66 instead of 0.2 for Mat\'ern and 0.1 for ours). On 3dRoad, a larger dataset where we outperform stationary kernels, ISMOE consistently achieved an NLL of 0.8 compared to our -0.25. This is better than Mat\'ern at 0.9, but worse than SVGP at 0.7. But ISMOE has an RMSE of 1, which is as good as always guessing 0 if the normalized data is Gaussian distributed, whereas ours achieves an RMSE of 0.18 and Mat\'ern achieves 0.1. On such a large dataset as 3droad, ISMOE also took 10 times as long as our method. A stationary kernel makes sense when we know that the relationship between data points depends on their relative position, regardless of their absolute location. But a stationary kernel also makes perfect sense if we do not know *how* the importance of relative position changes with respect to absolute location. So we do not see stationarity as a burdensome assumption made by the Mat\'ern kernel and others; it is as simple and reasonable an inductive bias as any. Given that we doubt the importance of stationarity, and we are not aware of ISMOE being widely used, and the method seems weaker than other methods we have compared against, we decided not to include it.
>
> Similarly, Trapp et al. (2020) seems to consistently underperform SVGP, which we consistently outperform.

---

> > ### Author Response · Authors · 2022-08-02
> > **Reply to concerns (part 2, mostly minor)**
> >
> > $>$ I think the high-dimensional kernel parameter $\theta$ could be somewhat hard to learn. How sensitive is the performance to the initial values of $\theta$? Also, the performance of BT is based on multiple initializations, which may not be fair to other competing methods.
> >
> > Depending on what you mean by "hard to learn" we might agree. It is probably very hard to find the optimal $\theta$. Even setting aside the fact that the loss surface is only piecewise smooth, one component of the loss surface (the log determinant part) has a Hessian with only one positive eigenvalue, if we recall correctly. But is it hard to learn a value of $\theta$ that achieves (at least) the kind of performance that we report? Our reported run times suggest: not very. Other methods may not use multiple initializations; maybe they rarely produce a different answer, or maybe it just isn't worth the time. But all methods are being judged by the same wall clock. We feel it would be too tangential to see how Wang et al.'s BBMM method could trade off run time against performance by using, for example, multiple initializations; we take it they decided multiple initializations were not worth it.
> >
> > **Minor points:**
> >
> > $>$ Could you please elaborate more on why the matrix inversion algorithm is called "approximate" inversion?
> >
> > Because for a matrix $M$, this algorithm can only calculate $(M + \lambda I)^{-1}$ for $\lambda > 0$. To do exact inversion, this calculation could be done repeatedly for smaller and smaller values of $\lambda$ until convergence. The run time analysis could be quite messy, because as $\lambda$ decreases, the floating point precision probably has to increase proportionally.
> >
> > $>$ Figure 2: Are those piecewise constant functions the posterior means of binary tree kernel GPs? I think a realization from the proposed GP should be piecewise smooth rather than piecewise constant.
> >
> > When $x^\textrm{raw}_1$ and $x^\textrm{raw}_2$ produce identical binary strings $x_1$ and $x_2$, $k(\cdot, x_1) = k(\cdot, x_2)$, so from the GP's perspective, $x^\textrm{raw}_1$ and $x^\textrm{raw}_2$ may as well be the same point, so the posterior mean must take the same value.

---

### Official Review · Reviewer_Lnbh · 2022-07-11

**Rating:** 6
**Confidence:** 4
**Soundness:** 3 good
**Presentation:** 3 good
**Contribution:** 3 good

**Summary:**

The manuscript proposes a GP covariance function to model multivariate piecewise constant functions so that computationally efficient inference and prediction becomes possible. The authors propose numerical primitives to compute matrix-vector multiplications, the inverse and log-determinant of the system matrix that scale linearly in the number of datapoints, the input dimension of the data and the inverse stepsize of the grid underlying the piecewise constant function both in terms of runtime and space.

**Questions:**

* The piecewise constant nature of the modeled function should be emphasized more clearly.
* The space/time footprint of the algorithms depends linearly (derivative of hyperparameters quadratically) on the data dimension and the stepsize of the underlying grid. This information should be presented more prominently.
* Also, the fact that the covariance is degenerate (low-dimensional) could be made more accessible.
* Given that in line 29, the manuscript puts a lot of emphasis on the variance estimate of inducing point methods, I would have expected a detailed comparison of the predictive variance of the proposed (degenerate) covariance with an exact GP with non-degenerate covariance and an inducing point method approximation.
* Can the numerical algorithms be cast such that also non-Gaussian likelihoods can be treated via expectation propagation, variational Bayes or Laplace approximation?
* An analysis, why "better [..] likelihoods" were achieved could be instructive. In particular, how does the likelihood depend on q? How are hyperparameters optimised
* There is no comparison to the exact model in terms of modeling accuracy.

**Limitations:**

Yes

**Strengths And Weaknesses:**

1) Clarity
The paper is mostly well written and the technical content is rather well accessible. The Algorithms 1+2 and would become simpler if consistent vector algebra was used instead of loops over indices.

2) Originality
The proposed covariance function and the derived algorithmical approaches seem to not have been explored before.

3) Significance
The method is (currently) specific to GP regression and could extend the range of GP models to high-dimensional data whenever piecewise constant functions and rank-deficient covariances are appropriate.

4) Reproducibility
As the code to run the experiments is included in the submission and the datasets are public, it should be rather simple to reproduce the results.

5) Empirical analysis
The manuscript contains a large number of numerical results gathered in a big table regarding marginal likelihood, prediction error and runtime. Some more results on more subtle aspects of the covariance function itsel would have been much appreciated.

---

> ### Author Response · Authors · 2022-08-02
> **Reply to concerns**
>
> $>$  Some more results on more subtle aspects of the covariance function itself would have been much appreciated
>
> We take it you are referring to other points you make below, but please let us know if there is anything else.
>
> $>$ Also, the fact that the covariance is degenerate (low-dimensional) could be made more accessible.
>
> In a sense, it is actually just as "infinite dimensional" as a Mat\'ern kernel or RBF kernel. For our kernel, the dimensionality of the Hilbert space is $2^q$, an admittedly finite number, but this means that the time complexity is $O(\log($dimension$))$. Traditional kernels are typically calculated with finite-precision floating point numbers. If one only uses 32-bit floating point numbers, no kernel is actually infinite dimensional. In practice, this is fine, because like ours, the dimensionality is exponential in the precision, while the run time is linear in the precision, so again the time complexity is $O(\log($dimension$))$. In comparison, the complexity of an inducing points method is $O($dimension$^2)$. Maybe it is clearest to say: unless binary strings are duplicated in the data, our kernel matrix has full rank.
>
> We never actually checked how often different data points are assigned the same binary string, but this comment inspired us to, because we do set $p$ much lower than 32. See our discussion in the general response.
>
> $>$ Can the numerical algorithms be cast such that also non-Gaussian likelihoods can be treated via expectation propagation, variational Bayes or Laplace approximation?
>
> We suspect our SROS representation for linear operators could be useful in many settings. It would be easy to extend our work to Student-t processes, for instance. We can imagine that there are very many settings where it would be useful to approximate a matrix in SROS form, perhaps including within some of the methods you mention, but we can't confirm this.
>
> $>$ An analysis, why "better [..] likelihoods" were achieved could be instructive. In particular, how does the likelihood depend on q? How are hyperparameters optimised
>
> We didn't actually optimize $q$. To avoid making the run time too large, we set $p$ to $\max(\lfloor 150/d\rfloor + 1, 8)$, and $q = pd$. We have added a sensitivity analysis on performance under different choices of $p$, which we describe in our response to all reviewers.
>
> For the other hyperparameter $\lambda$, if the last weight in the weight vector increases by $x$, and $\lambda$ decreases by $x$, the resulting kernel is the same. Since the last weight in the weight vector is optimized, it doesn't really matter what $\lambda$ is, except for the fact that the weight must be non-negative. So our hyperparameter $\lambda$ is really just a lower bound on the "whiteness" of the kernel. We simply set $\lambda = 1/n$ out of a concern for overfitting in the low-data regime, but if we wanted whiteness optimized, we would set it to the lowest numerically stable value, and let the last weight of the weight vector do the work.
>
> $>$ There is no comparison to the exact model in terms of modeling accuracy.
>
> Our method is not an approximation of some other method, so there is no such thing as an exact model that we fail or manage to replicate.
>
> $>$ Given that in line 29, the manuscript puts a lot of emphasis on the variance estimate of inducing point methods, I would have expected a detailed comparison of the predictive variance of the proposed (degenerate) covariance with an exact GP with non-degenerate covariance and an inducing point method approximation.
>
> We're not sure how this comment would change in light of the observation that the covariance of our method is not, in fact, degenerate, and our method is not an approximation of some "exact" alternative.
>
> **Minor points:**
>
> $>$ The Algorithms 1+2 and would become simpler if consistent vector algebra was used instead of loops over indices.
>
> With the exception of line 7 in Algorithm 2, there isn't standard vector algebra notation for any of these operations, because the indices on the left and right hand sides typically don't all match.
>
> $>$ The piecewise constant nature of the modeled function should be emphasized more clearly.
>
> $>$ The space/time footprint of the algorithms depends linearly (derivative of hyperparameters quadratically) on the data dimension and the stepsize of the underlying grid. This information should be presented more prominently.
>
> We now have presented this more prominently, expect we phrase it a bit differently because a larger dimension often makes small stepsize less necessary. See CTSlice, a high-dimensional dataset where we perform the best, and where we set $p=1$.

---

### Author Response · Authors · 2022-08-02
**For all reviewers**

Thank you so much for your interest, your scrutiny, and your thoughts. We have updated our draft, and we colored teal some revisions that you might want to check.

In our revision, two key additions are:

1. A sensitivity analysis to the choice of $p$ (see Appendix D, Table 2). Unsurprisingly, the RMSE and NLL decrease monotonically as $p$ increases for the tested values of $p=2,4,8$. Wall time also increases monotonically with $p$. Thus, the choice of $p$ should be informed by resource constraints, but larger is better. That makes the heuristic rule that we use to set $p$ in our experiments a reasonable choice.

2. A simple improvement in the setting where many different data points are assigned the same binary string (inspired by a comment by Reviewer Lnbh). In most cases, there are few to no such collisions, meaning our kernel matrix is full rank or very nearly so. Not so surprisingly, although it did surprise us, when $p$ is small, some datasets give many collisions. So we discovered a way to improve our performance, and we have added these new results in an appendix. First, using our existing rule to set $p$, we check how many collisions are in the training data, and if there are lots, we do two things. The first comes from an observation about rescaling. Recall that we rescale all our data to be in $[0, 1]^d$. In some datasets, *cough* Buzz *cough*, very few large outliers cause the bulk of the data to be squished into a very small region, and then end up in the same bucket when mapped to binary strings. So in datasets with a high collision rate, we apply a strictly increasing, piecewise linear transformation to the data, mapping the $k$th percentile of each dimension to $k/100$, for several values of $k$. The second is to increase $p$ to see if it helps reduce collisions. We did this for the three datasets with a meaningful collision rate, and put those results in an appendix. On 2/3 of these datasets, BTE with these improvements achieves the lowest NLL and RMSE across all datasets. Importantly, these two changes (the transformation and the increase in precision) can be applied when necessary: the \% of unique strings can be checked simply on the training set. We provide these results in Appendix D.

---

### Meta-Review · Area_Chair_PLud · 2022-08-24

**Recommendation:** Accept
**Confidence:** Less certain

**Metareview:**

Overall, reviews on this paper are positive. The paper presents a novel binary-tree based kernel which yields a structured kernel matrix that allows for near linear time GP inference. The approach is interesting theoretically and effective in practice. It is a valuable alternative to inducing point methods or other approaches that start with a standard kernel and then speed up inference via kernel matrix approximation.

The reviewers have made several suggestions to improve the presentation and empirical evaluation that I hope the authors will incorporate in the final version. Two important points that came up after author-reviewer discussion that I would like to point out are:

1. The matrix structure that arises from the binary tree kernel (which the authors call SROS) is exactly a classic hierarchical matrix structure, with rank-1 blocks. See e.g., https://www.mis.mpg.de/scicomp/Fulltext/WS_HMatrices.pdf. These matrices are extremely well studied in scientific computing, and it is well known that they admit fast linear system solvers, etc. it is surprising that the authors don't make a connection here -- presumably they are not aware of the prior literature. We feel that adding a discussion of the connection to hierarchical matrices is critical for the camera ready version. This would include a discussion of how the algorithms presented in the paper relate to existing algorithms for these matrices.

2. The discussion of 'recall' in the introduction needs a significant overhaul or needs to be removed. In particular, this discussion is not backed up with any evidence or any citations. None of the empirical results of the paper address this claimed advantage over inducing point methods. Citations need to be a added to this area of discussion, and empirical results backing it up need to be added. Otherwise, it should be removed from the paper.


Thanks!

**Award:**

No

---

### Decision · Program_Chairs · 2022-09-14

Accept